



# Classifying extratropical cyclones and their impact on Finland's electricity grid: Insights from 92 damaging windstorms

Ilona Láng-Ritter[1], Terhi K. Laurila[1], Antti Mäkelä[1], Hilppa Gregow[1], and Victoria A. Sinclair[2]

[1]Finnish Meteorological Institute,P.O. Box 503, 00101 Helsinki, Finland
[2]Institute for Atmospheric and Earth System Research/Physics, Faculty of Science, P.O. Box 64, University of Helsinki, 00014 Helsinki, Finland

**Correspondence:** Ilona Láng-Ritter (ilona.lang-ritter@fmi.fi)

**Abstract.**

This study investigates the impacts of extratropical cyclones on Finland's electricity grids, focusing on 92 significant windstorms from 2005 to 2018. We present a classification method for extratropical cyclones based on the arrival location and direction. Rather than using meteorological criteria to identify windstorms, we select them based on
their impacts to reach a more targeted understanding of windstorm impacts compared to traditional approaches. Key findings indicate that southwest-originating windstorms cause the most damage in total, while northwesterly windstorms lead individually to the highest average outages. The largest impacts occur when a windstorm moves across the northern part of a country, from the northwest to east, with the strongest wind gusts concentrated on the southern side of the low-pressure center, on highly populated regions. From the meteorological characteristics of
windstorms, the most relevant for grid damage besides the wind gust speed is the extent and spatial distribution of wind gusts. The seasonal analysis shows that windstorms are more frequent and damaging in autumn and winter, but even weaker wind speeds during summer can cause significant damage. Factors such as soil frost influence the severity of windstorm damage, highlighting the importance of expanding research to include environmental and geographical aspects.

Keywords: Extratropical cyclone, windstorm, electricity grid, power outages, resilience, impact forecasting, wind risk.

## 1 Introduction

Extratropical cyclones are among the most significant natural hazards in the mid-latitudes, driving daily weather
fluctuations and potentially causing substantial impacts (Browning, 2004). These large-scale low-pressure systems,



associated with fronts where cold and warm air masses meet, can become particularly impactful if they intensify into windstorms. Intense extratropical cyclones pose considerable risks to society (Wernli et al., 2002) and forests (Schelhaas et al., 2003; Schelhaas, 2008), particularly due to extreme winds, which notably impact critical infrastructure such as power distribution networks. Disruptions to distribution grids can lead to widespread power outages,
significantly impacting residences and essential services (Panteli and Mancarella, 2017).

Climate change is expected to influence the frequency and characteristics of extratropical cyclones and windstorms. Although projections for wind speeds associated with extratropical cyclones remain uncertain (Christensen et al., 2013; Masson-Delmotte et al., 2018; Priestley and Catto, 2022), some trends are generally agreed upon. The total number of extratropical cyclones is expected to slightly decrease (Bengtsson et al., 2006; Catto et al., 2011; Zappa
et al., 2013; Chang et al., 2013; Sinclair et al., 2020), and while the mean intensity of extratropical cyclones is not anticipated to change significantly, the most extreme cyclones are expected to become even more extreme (Bengtsson et al., 2006; Ulbrich et al., 2009; Zappa et al., 2013). In Finland and northern Europe, the climate projections suggest only minimal changes in mean (Ruosteenoja and Jylhä, 2022) and extreme wind speeds with a slight increase of up to 2.5% in extreme winds during autumn by the late 21st century (Ruosteenoja et al., 2019).
While the overall windstorm and wind climate in Finland and Northern Europe may not change significantly, the potential for damaging windstorms could still increase. This is due to factors such as the expansion of areas impacted by intense winds during extreme windstorms by up to 40% (Priestley and Catto, 2022) and an increase in cyclone wind speeds near the warm sector by up to 3.5 m s$^{-1}$ (Sinclair et al., 2020). Additionally, climate change is expected to intensify tropical cyclones and shift them eastward, increasing the risk that they can undergo extratropical
transition and impact northern Europe more frequently than before (Haarsma, 2021).

Extratropical cyclones have been the focus of various classification efforts based on meteorological features. Classifications can help explain the physical mechanisms driving societal impacts (Catto, 2016). Early methods relied on simple conceptual models based on cyclone structure (e.g., Bjerknes and Solberg (1922); Shapiro and Keyser (1990)) or life cycle characteristics. The most common approaches classify cyclones by their intensity, with studies
using metrics such as maximum 850 hPa relative vorticity or minimum sea level pressure (MSLP) (Catto et al., 2010; Sinclair et al., 2020) or rapid deepening rates (e.g., a 24-hour pressure drop, also known as meteorological bombs (Sanders and Gyakum, 1980)). A recent analysis classified cyclones into clusters based on multiple intensity measures—850 hPa vorticity, wind speed, wind footprint, precipitation, and a storm severity index—highlighting variability in intensity and life cycle characteristics (Cornér et al., 2024). Other classifications focus on cyclone
origins, such as transitioning cyclones (Klein et al., 2000), Mediterranean cyclones (Campins et al., 2000), or polar lows (Rasmussen and Turner, 2003). Since wind is often the most damaging feature of extratropical cyclones, classifications have utilized statistical methods to define extreme events based on high wind gust percentiles (90th, 98th, or 99.5th) (Laurila et al., 2021a; Klawa and Ulbrich, 2003; Nissen et al., 2010). Societal impact assessments are typically conducted following the meteorological classification of hazards (e.g., Klawa and Ulbrich (2003)). However,





studies rarely classify or filter events primarily based on their societal impacts, which can result in overlooking the most critical aspect: the societal consequences.

The IPCC risk framework (Campos et al., 2014) describes how the physical climate hazard, exposure, and vulnerability interact to produce risk. Power transmission and distribution networks, especially in regions with overhead lines, are highly vulnerable to windstorms. The risk of windstorms is widely present in Europe, with over 10 million

kilometers of power lines (Mandatova and Lorenz, 2019), and windstorms indeed often cause significant outages, particularly in low- and medium-voltage regional grids. In forested regions such as Finland, where 70% of the land is covered by forest (Venäläinen et al., 2020), the risk of power outages is common due to trees falling on power lines. For example, 46% of transmission faults in Finland from 2010 to 2018 were due to windstorms, with peak years reaching 69% (Tervo et al., 2021). Measures like underground cabling and relocating lines are being increas-

ingly implemented to mitigate the risk and improve resilience, though they come at high costs (Nurmi et al., 2019). Balancing effective adaptation with cost-efficiency is essential for mitigating outage risks (Jasiūnas et al., 2023a).

This study examines the impacts of extratropical cyclones and windstorms on Finland's electricity grids, focusing on the 92 most significant windstorms in terms of their impact from 2005 to 2018. We aim to provide a more targeted understanding of windstorm impacts compared to traditional classifications based purely on meteorological

parameters. The goal is to enhance preparedness for future windstorms and wind risks, and offer guidance for electric grid resilience and emergency response planning in Finland or similar regions. We address these objectives by developing a novel classification for all extratropical cyclones based on their arrival location and direction, and climatological locations of the strongest wind gusts. We then identify windstorms through their impact, such as power outages, instead of meteorological features. Furthermore, we compare the meteorological characteristics of

windstorms to extratropical cyclones by class, determine windstorm-related meteorological properties (e.g., storm tracks), and quantify how the impacts vary depending on the type of windstorm and its meteorological characteristics. We further investigate how the impacts vary by region and season across different windstorm classes.

The paper is structured as follows: Sect. 2 introduces the meteorological and power outage data, Sect. 3 outlines the methodology for classifying and tracking windstorms, Sect. 4 presents the results, showing windstorm class

characteristics, impacts on the power grid, and the seasonality of damaging windstorms, and Sect. 5 concludes the study.

## 2  Data

### 2.1  Meteorological data: wind gust observations and ERA5 reanalysis

In this section, we describe the wind gust observations and ERA5 reanalysis data used to identify typical char-

acteristics of extratropical cyclones and windstorms. We utilized multiple data sources to better understand the relationship between wind gusts and windstorm impacts in Finland.



The SYNOP observation data of Finnish Meteorological Institute (Finnish Meteorological Institute, 2020) are quality-controlled meteorological measurements, and we utilized 10-meter wind gusts from 180 observation stations for the period 2005–2018. The wind gust observations were used in Section 4.4.1 to examine the most significant factors affecting the electrical grid by analyzing the correlation between key electrical grid parameters and meteorological wind gust data.

ERA5 reanalysis (Hersbach et al., 2020) is the latest reanalysis (i.e. combination of weather observations and past short-term forecasts (ECMWF, 2020)) produced by the European Centre for Medium Range Weather Forecasts (ECMWF). It has a horizontal resolution of 31 km and 137 vertical levels with the top level at approximately 80 km altitude. The meteorological parameters included in ERA5 are available at hourly resolution from 1940 to the present. In this study, we use the mean sea level pressure (hereafter MSLP), and maximum wind gust (maximum 3-second wind at 10-m height), since these parameters are often referred to as a measure of the intensity of an extratropical cyclone. The 3 s wind gust is computed every time step (1 h for ERA5) and the maximum value is kept since the last post-processing period (ECMWF, 2022). Maximum wind gusts are a dominant factor for wind impacts: trees tend to fall (uproot or break) especially due to turbulence and wind gusts of extratropical cyclones, causing a significant part of forest damage and electrical grid failures in forested areas (Gardiner et al., 2013).

ERA5 reanalysis provides a spatially uniform dataset compared to the in situ weather observations. The wind gusts in ERA5 are parametrized and calculated as a sum of three terms: 10-m wind speed, the turbulent gust term, and the convective gust term (ECMWF, 2016). Ramon et al. (2019) evaluated ERA5 wind speeds on daily, seasonal, and decadal scale comparing wind observations from 77 globally placed tall-tower sites with reanalysis wind speeds, and discovered that ERA5 provides the best correlation with the observed wind speeds especially on the daily scale. ERA5 data is known to overestimate surface wind gust speeds in some areas (ECMWF, 2019), however, these do not fall into the domain of this study. ERA5 is a viable alternative to in situ observations of wind gusts since it provides a spatially uniform dataset covering a longer period in higher temporal resolution than wind gust observations. Moreover, wind gust observations can be inhomogenous due to the differences in measurement instruments, observation frequencies, or observation station locations (Feser et al., 2015). In Finland, for example Laapas and Venäläinen (2017) homogenized wind observations with statistical tools and found that up to 95 % of the analysed wind speed time series were inhomogenous. In this paper, we use ERA5 wind parameters for defining characteristics of extratropical cyclones and windstorms in Section 4.2. By combining reanalysis and observations, we aim to capture both the broad-scale atmospheric conditions and localized variations that influence windstorm impacts in Finland.

## 2.2 Impact data: Power outages

In this section we describe the power outage data, which was used to connect the impacts with the meteorological features of the windstorms and extratropical cyclones. The analysis period was determined by the temporal coverage of the power outage data: a 14-year period from 2005 to 2018. The outage data in Finland is recorded by multiple





independent distribution system operators (hereafter DSOs). The scattered outage data is collected and merged at national level for further yearly statistical analysis by the consulting company ENEASE Oy (ENEASE Oy, 2017). Finally, the yearly statistics are published by Finnish Energy (2021), which is the union of DSOs. ENEASE Oy and Finnish Energy can share data with third parties as long as the identity of DSOs is not revealed. To ensure

anonymity and fair competition for the companies, the outage data is provided as aggregates for areas comprising a minimum of six DSOs, shown in Figure 1. This aggregation results in the data being provided for five geographical areas that are based on the operating areas of DSOs and the boundaries of the provinces of Finland. The dataset has been previously utilized also by Tervo et al. (2021); Jasiūnas et al. (2023a, b, 2024) and Haakana et al. (2024) to develop impact models to predict power outages induced by extratropical cyclones and to investigate the resilience

of the electricity distribution system.

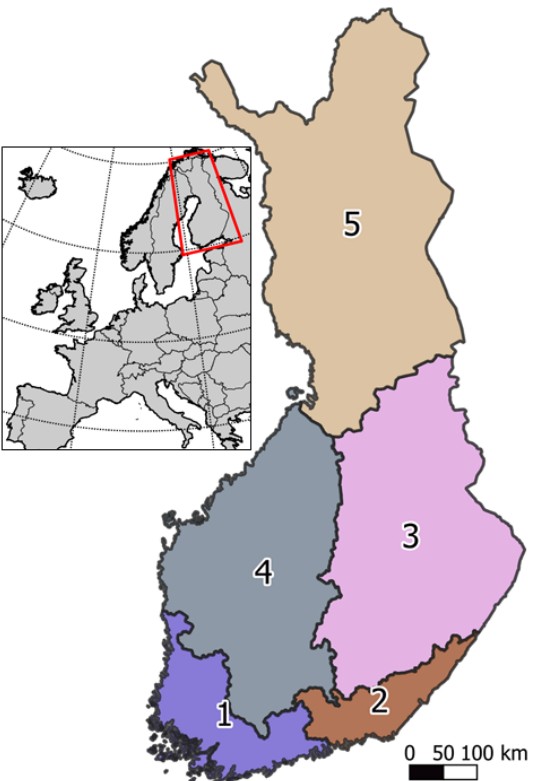

**Figure 1.** The aggregation regions of the power outage dataset. Later, when we refer to regions 1-5, we call them as follows: Area 1 = Southwest region (including the capital Helsinki), Area 2 = Southeast region, Area 3 = East region, Area 4 = West region, and Area 5 = North region

Each power outage in the outage dataset is labelled according to its most plausible cause. The original dataset contains labels for meteorological causes, such as 'thunder', 'snow load' or 'wind and storm', but also non-meteorological





causes like 'animals' or 'unknown' are included. The cause labels are assigned by the engineers repairing the faults in the field and can therefore contain occasional errors, e.g. mistaking the cause 'wind and storm' for the cause

'thunder'.

Among the impact attributes recorded in the dataset, we focus on the following two categories: number of delivery points without electricity (hereafter NDP) and faults. NDP represent roughly the number of households without electricity, and 'faults' describe the amount of faults in the electrical grid, i.e. one fault in a critical place of the grid structure can result in power cuts in several households. We consider NDP more relevant and meaningful than faults

for scientists and forecasters, and therefore mainly use this category in our analysis. However, the total number of faults may be a more important measure of impacts for DSOs. Therefore we include them in the correlation analysis, in Section 4.4.1.

The original outage data consists of a total of 581 430 faults (number of data rows) for the period 2005-2018. We have removed duplicates, faults with unrealistic duration and time values (negative duration, or non-existing dates),

and empty data entries, resulting in a total of 571 634 faults. Among these, the number of faults labeled with the cause 'wind and storm' is 238 869, representing 42 % of the filtered faults. The sum of NDP in the 'wind and storm' category was 31 121 052.

## 3    Methods

To strengthen the understanding of the dependency between windstorms and their impacts, we developed a novel

classification method for extratropical cyclones, based on their propagation track and arrival location to the area of interest, and the related impacts. In the following subsections we describe the three steps of the method in detail: 1) Identifying the extratropical cyclone tracks and characteristics (Section 3.1) 2) Identifying the most impactful windstorms based on power outages 3.2 3) Classifying the extratropical cyclones and windstorms (Section 3.3).

### 3.1    Extratropical cyclone tracking and characteristics

An extratropical cyclone track consists of the latitude and longitude coordinates of the low-pressure centre at each point along its propagation route. In this study, the extratropical cyclone tracking was performed by applying the TRACK method (Hodges, 1994, 1995, 1999) to the ERA5 MSLP data from 2005–2018. We tracked all extratropical cyclones in the Northern Hemisphere based on the localised minimum MSLP at 3-hourly temporal resolution and regridded to T63 resolution (around 180 km). The interpolation to coarser resolution was made to reduce noise in the

MSLP fields and to only select synoptic-scale extratropical cyclones. From these tracks, we first excluded short-lived (lifetime less than one day) and stationary (moving less than 500 km during their lifetime) tracks. Second, we found the value of the native i.e. high-resolution MSLP at each point along each track. Third, we identified the maximum 10-m wind gust at the native resolution within a 6 degree geodesic radius of the cyclone centre. The 6 degree radius was chosen as Laurila et al. (2021a) found that the maximum wind gusts associated with windstorms are commonly





observed within that radius. To narrow down the number of considered extratropical cyclones to those potentially affecting Finland we only retain the tracks that passed through the domain of 0–60°E, 50–75°N. This resulted in a total of 3304 extratropical cyclones in the 14 year period.

After identifying the extratropical cyclone tracks, various meteorological characteristics were derived from ERA5 data. The lifetime of each extratropical cyclone was calculated over its entire track whereas the minimum MSLP, mean
propagation speed, maximum deepening rate, maximum wind gusts, and percentage of ERA5 grid cells exceeding 21 m s$^{-1}$ (official storm definition in Finland is when 10-min average wind speed exceeds 21 m s$^{-1}$ (FMI, 2022)) were calculated only inside the Finnish domain (i.e. the red box in Figure 1). The mean propagation speed of each extratropical cyclone was obtained by averaging the propagation speeds of all time steps. The deepening rate describes how the minimum MSLP of the track changes in time and it can be either negative (cyclone is deepening) or positive (cyclone is decaying). We calculated the deepening rate for each 3-hour time interval separately and then
defined the maximum deepening rate to be the biggest negative change in MSLP in 3 hours. The percentage of ERA5 grid cells (31km x 31km) with wind gusts exceeding 21 m s$^{-1}$ was calculated by defining a percentage of how many grids inside the domain of Finland have gust values exceeding 21 m s$^{-1}$ while the track is inside the domain of Finland (20.0–31.4°E, 59.5–70.1°N (i.e. the rectangle box in Figure 1 and 2)).

### 3.2 Identifying the most impactful extratropical cyclones based on power outages

In this study, we are interested especially in the most impactful extratropical cyclones (i.e. windstorms). Due to the high temporal resolution of the power outage data (one second), see Section 2.2, it is relatively straightforward to connect the power outages to specific meteorological events (e.g. wide power outage cases caused by extratropical cyclones). We further filtered the 'wind and storm'-related outages to retain only those occurring on days when
the overall NDP amount exceeded 50 000. Using this impact-based threshold to filter the windstorms, we were able to identify 119 days with the highest reported impacts. We further validated these 119 days date-by-date, finally excluding 17 dates where either 1) the same windstorm caused power cuts during two consecutive days, 2) no extratropical cyclone tracks were found, or 3) outages were caused by a small-scale convective storm. This resulted in 92 extratropical cyclone cases, which are referred to throughout this document as "windstorms" to be investigated
more thoroughly in this study.

### 3.3 Extratropical cyclone classification

In this section we describe the classification that is applied to all extratropical cyclones (including windstorms). The extratropical cyclone classification consists of two main steps: the cyclones are classified based on their 1) arrival location and 2) arrival direction to Finland.





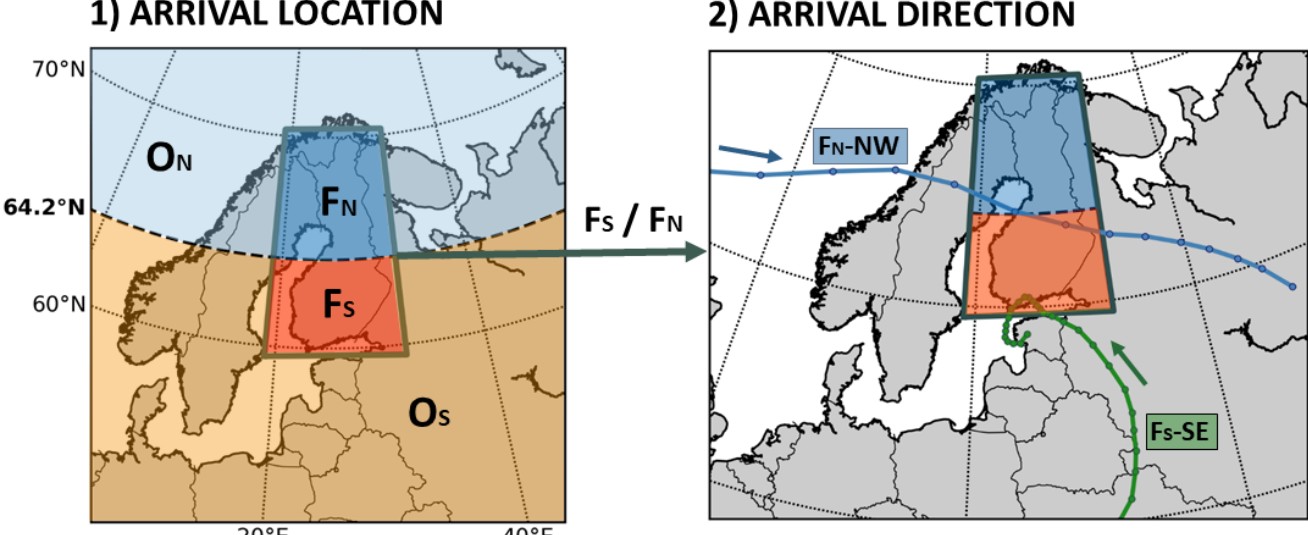

**Figure 2.** The two steps of the extratropical cyclone classification method. 1) The first part of the class name is defined according to the arrival location of the extratropical cyclone. If the low-pressure center does not pass directly over Finland at all, it is classified as class O (Outside Finland). The O class extratropical cyclone gets an additional letter N (North, light blue area) or S (South, orange area) depending on which side of the latitudinal center (64.2°N) of Finland the storm arrives from. In case the extratropical cyclone passes over Finland, it is classified to class $F_N$ (Northern Finland, dark blue box) or class $F_S$ (Southern Finland, red box). 2) In the second part of the classification, $F_N$ and $F_S$ extratropical cyclones are further classified based on their arrival direction. The arrival direction is calculated as an average of 3-hour propagation directions over the previous 12 hours of the extratropical cyclone track before entering the box. Thus, 0–90° is classified as NE class, 90–180° as SE class, 180–270° as SW class, and 270–360° as NW class.

Rather than using real geographical borders, we defined the domain of Finland within 20.0–31.4°E, 59.5–70.1°N (i.e. the box rectangle in Figure 1 and 2). First, we identify whether the center of the extratropical cyclone passes over Southern (class $F_S$) or Northern Finland (class $F_N$), or whether it does not pass directly across Finland at all (class O, Outside of Finland). Class $F_S$ contains cyclones which low-pressure centers are located south of the latitudinal center of Finland (64.2°N, Figure 2) and class $F_N$ the ones north of this line. To further classify the class

O cases, we first found the closest coordinate of the cyclone track in comparison to the longitudinal center of the Finland domain (25°E). Then, we checked whether the latitude of the storm track at that point is north or south of the latitudinal center of Finland (64.2°N) and defined an additional definition as class $O_N$ to northerly and class $O_S$ to southerly cyclones. For example, an extratropical cyclone that never passes through the domain of Finland, and, when it is closest to Finland is located farther north than 64.2°N, is classified as an $O_N$ extratropical cyclone

or windstorm. In the second step of the classification, the direction from which the extratropical cyclones of class $F_S$ and $F_N$ approaches Finland was determined by calculating the mean propagation direction of the previous 12 hours





before entering the domain (Figure 2). Based on this average propagation direction, the extratropical cyclone class gets an additional definition: 0–90° is classified as NE class, 90–180° as SE class, 180–270° as SW class, and 270–360° as NW class. Thus, for example, an extratropical cyclone that arrives to northern Finland (north of 64.2°N) and from 295° direction is classified to $F_N$-NW class (e.g. the blue track in Figure 2).

## 4   Results

In the previous section we classified windstorms and extratropical cyclones based on their impacts, and arrival location and direction in respect to Finland. By connecting the windstorms with impacts, we describe the differences between the impacts of different windstorm classes in the following sections. In Sections 4.1, 4.2 and 4.3 we describe the meteorological characteristics of the windstorms and extratropical cyclones. In Section 4.4 we focus on the windstorm impacts on the electric grid in Finland.

### 4.1   Extratropical cyclone and windstorm classes

Figure 3 represents the distribution of the number of extratropical cyclones and windstorms, and the ratio of windstorms/extratropical cyclones in each class. During 2005–2018, in total 3304 extratropical cyclone tracks were found, of which 713 occurred within the domain of Finland ($F_N$ and $F_S$ classes). Thus, every week four to five extratropical cyclones occurred in total, and in the domain of Finland on average one extratropical cyclone passed by per week. Of all the identified extratropical cyclones, 92 were classified as 'windstorms' based on their impacts in Finland, i.e. they caused a minimum of 50,000 NDP without electricity (Section 3.1). Windstorms occurred on average only six times per year. Almost 3% of the extratropical cyclones were classified as windstorms (classes O and $F_N$ and $F_S$). Of the extratropical cyclones, which passed over Finland (class F), almost 10 % were windstorms.

Based on our results, the extratropical cyclones outside of the domain of Finland (class O, Figure 3) are overall the most common type of extratropical cyclone. However, quite a small share of class O cyclones actually impact the electrical network in Finland (around 2%). This is most likely because although they are inside of our defined larger domain, they either never arrive in the very close vicinity of Finland and their highest wind gusts are located far from Finland. Among the windstorms analysed, the share of class O is however significant: these represent a total of 23, which equals to 25 % of all the damaging windstorms. Although the extratropical cyclones in class $O_S$ are more common (1335 cyclones) than in class $O_N$ (985 cyclones), Finland is impacted more often by the class $O_N$ (14 windstorms) than the $O_S$ windstorms (9 windstorms). This is most likely due to the strongest wind gusts being located on the southern side of the windstorm track. In $O_N$ windstorms, the strong winds reach Finland especially in the Northern and Western parts because of their typical route through the Northern Atlantic and Barents sea. In contrast, for $O_S$ windstorms, the strong winds often remain outside of the domain of Finland, and when they do extend over Finland, they rather impact Southern or South-Eastern Finland.





**Figure 3.** Number of all extratropical cyclones and windstorms (2005–2018), and the share of windstorms of the total number of extratropical cyclones in each class. Light green bars represent the total number of all extratropical cyclones and dark green bars the windstorms (extratropical cyclones causing at least 50,000 NDP without electricity) within the domain of Finland (0–60°E, 50–75°N).

In classes where the extratropical cyclones pass over Finland ($F_N$ and $F_S$, see Figure 3), a South-Westerly propagation direction was the most frequent. In total we identified 263 extratropical cyclones in class $F_S$-SW, of which 19 were windstorms, and 147 in class $F_N$-SW with 26 windstorms. It comes as no surprise that a majority of windstorms (around 70%) arrived from westerly directions, (south-west 49%, north-west 21%) because of Finland's location in relation to typical storm tracks in Northern Europe. In general, the windstorms with a Northern storm track (class $F_N$ and $O_N$ windstorms) were more often damaging than the ones having a southerly route concerning Finland ($F_S$ and $O_S$ windstorms). Among all classes, $F_N$-SW had the highest ratio of windstorms in relation to all extratropical



cyclones (17.7%). North-West is also a relatively frequent arrival direction of extratropical cyclones passing over Finland. As Figure 3 shows, there were in total 139 class $F_N$-NW extratropical cyclones (14 windstorms) and 100 class $F_S$-NW extratropical cyclones (5 windstorms). The most seldom arrival directions for extratropical cyclones and windstorms in Finland are NE and SE. There were 44 extratropical cyclones in class $F_S$-SE of which 4 were classified as windstorms. The rarest class was $F_N$-SE (6 extratropical cyclones, 1 windstorm). The North-Eastern
arrival direction was very rare and there were no windstorms among the extratropical cyclones in classes $F_N$-NE (7 extratropical cyclones) and $F_S$-NE (7 extratropical cyclones).

## 4.2 Extratropical cyclone and windstorm characteristics

The characteristics of extratropical cyclones and windstorms can indicate also their damaging potential. Therefore, first we selected some of the characteristics known to be connected to the impacts of the cyclones. Second, we
compared the characteristics of all extratropical cyclones (713) with the 69 windstorms which passed over Finland (class F) (Figure 4). In the characteristic comparison, we only considered classes F because a comparison with classes O would not be fair. This is due to the large size of the bounding box including numerous cyclone tracks in class O that do not affect the domain of Finland at all.

As Figure 4a shows, the lifetime of the extratropical cyclones can vary from less than 50 hours to up to over
500 hours i.e. from 2-21 days. The median lifetime across all classes ranges from 70 to 150 hours, with $F_S$-NW windstorms having the shortest lifetime and $F_S$-SW windstorms the longest. When comparing the extratropical cyclones and windstorms, there does not seem to be a consistent difference between the lifetimes of the windstorms and extratropical cyclones, which indicates that the lifetime of the extratropical cyclone is likely not a strong predictor for the eventual amount of damage. On the contrary, the minimum MSLP of the extratropical cyclones
and windstorms differ very clearly (Figure 4b). When comparing the minimum MSLP of the extratropical cyclones and windstorms, we observed that in each of the classes, the median of the minimum MSLP of the windstorms is 10–20hPa lower than that of the extratropical cyclones. In addition, the minimum MSLP also differs between the classes. $F_N$-NW and $F_S$-NW have the lowest medians of minimum MSLP, reaching below 980 hPa. On the contrary, class $F_S$-SE windstorms have the highest median of minimum MSLP which indicates they are not so intense extratropical
cyclones. When considering the propagation speed of the extratropical cyclones and windstorms, there are differences between the classes (Figure 4c). The $F_S$-SE windstorms have the slowest propagation speed, with a median of 20 km h$^{-1}$. On the other hand, $F_N$-NW windstorms propagate the fastest, with a median propagation speed of 75 km h$^{-1}$. The propagation speed is a significant characteristic for two main reasons: 1) slow windstorm propagation speeds indicate that the strong wind gusts influence the same area for a long time, thus the damage accumulate, or 2) fast
windstorm propagation speeds suggest that the storm has intense meteorological characteristics, such as extreme maximum wind gusts, leading to significant damage.

The maximum deepening rate of the cyclone also suggests a possibility of an intense windstorm, in case the development is very rapid. The median of the maximum deepening rates of the windstorms vary between -0.5 and





0.3 hPa h$^{-1}$, and the most extreme deepening rates are between -1.5 ( F$_S$-SE) and -0.8 hPa h$^{-1}$ (F$_N$-SW). Previous
studies show that rapidly deepening extratropical cyclones that move for instance across central Finland, presents a
significant risk of wind damage, particularly for the southern half of the country (Valta et al., 2019; Gregow et al.,
2020).

Maximum wind gusts of the windstorms are often considered the most important indicator connected to the
damage for electrical grid. The maximum wind gusts in different classes are presented in Figure 4e. In the most
extreme F$_S$-NW windstorms, the median wind gust speed is 32 ms$^{-1}$. In contrast, the lowest wind gust median 24
ms$^{-1}$ was found in class F$_S$-SE windstorms. The maximum wind gusts are clearly higher in windstorms than in all
extratropical cyclones.

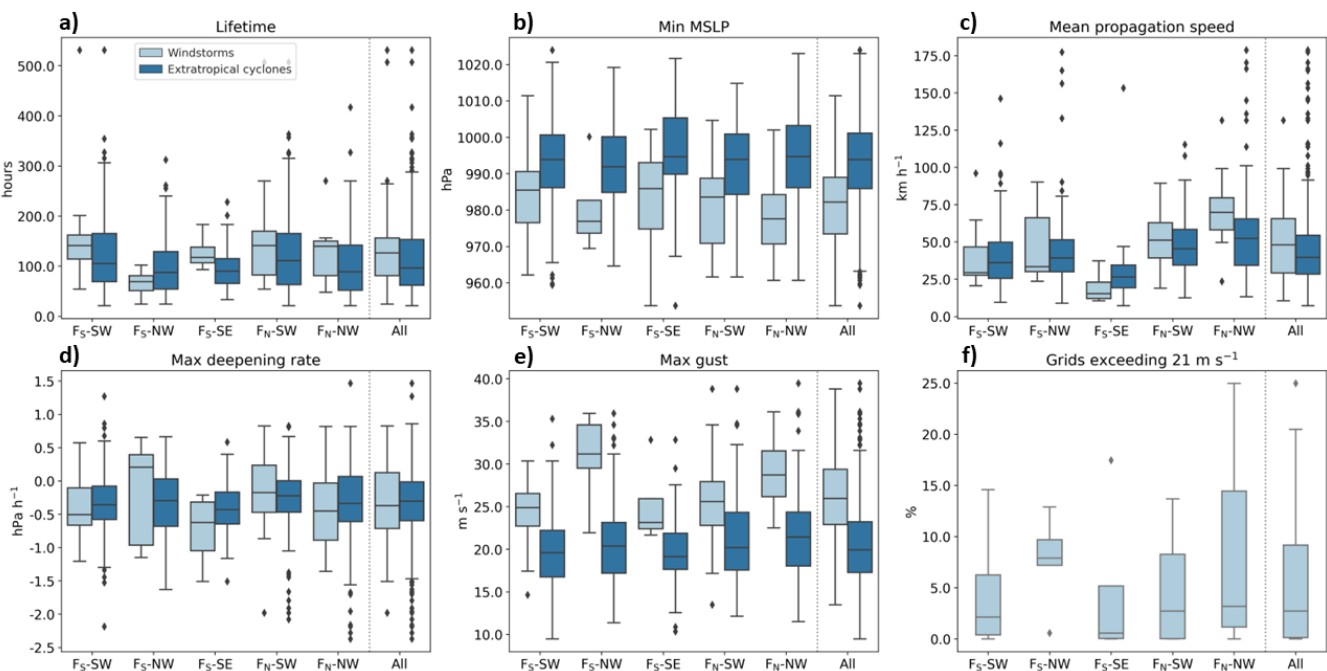

**Figure 4.** Extratropical cyclone characteristics inside the domain of Finland in each class. Light blue boxes represent wind-
storms and dark blue boxes extratropical cyclones. The following characteristics are shown: a) track lifetime, b) minimum
mean sea level pressure, c) mean propagation speed, d) maximum deepening rate, e) maximum wind gust, and f) percentage
of ERA5 grid cells (31x31km) with wind gusts exceeding 21 m s$^{-1}$ in the Finnish domain. The 'All' column on the right
displays the characteristics of all examined windstorms (69) and extratropical cyclones (713) for comparison.

The amount of windstorm damage can be also influenced by the size of the area affected by strong wind gusts.
Thus, we compared the areas of strong wind gusts in each windstorm class by identifying the percentage of ERA5





grid cells inside the domain of Finland with gust values exceeding 21 ms$^{-1}$ while the windstorm track is inside the domain (Figure 4f). The median of the percentage of ERA5 grid cells exceeding 21 ms$^{-1}$ varies between 2 to 9 %, however, in the most extreme situations the gusts of over 21 ms$^{-1}$ can cover 15 to 25 % of the domain of Finland. The smallest median of strong wind gust areas is found in class $F_S$-SE. This result is in line e.g. with the study of Laurila et al. (2021a) who show that the strongest gusts typically occur on the southern side of the storm track and the Southern arrival direction of $F_S$-SE windstorms (described more in details in Section 4.3). This means that the $F_S$-SE windstorms often influence only a small area in Southern Finland (regional analysis in more detail in Section 4.4), because the strongest wind gusts usually stay on the southern side of Finland unless the storm track moves more north over the country. In contrast, the windstorms having more northern storm tracks have slightly higher coverage of strong wind gusts. When a large part of Finland is influenced by the warm sector and the cold front of the windstorm, there is also wider coverage of the grid cells exceeding 21 ms$^{-1}$ threshold. For example, $F_S$-NW has the highest median of the strong wind gust area (9%) and $F_N$-NW has the highest individual value of windstorm's wind gust coverage (25 %). Generally speaking, the medians of strong wind gust areas do not appear to be a large number compared to the overall area, however, also the location of the strong wind gusts can impact the amount of damage (more thoroughly explained in 4.4). Generally, Northern Finland is very sparsely populated, thus most of the electrical grids in Finland are located in southern and central parts of Finland and those parts are also more vulnerable for strong winds. On the other hand, in the South, the electrical cables are more often under the ground which makes the electrical grid more resistant against windstorms.

### 4.3 Tracks, genesis and lysis locations of the windstorms

To be able to understand in more depth the windstorm characteristics, we studied the tracks of the windstorms and in particular where the development of the storms starts (genesis) and ends (lysis). We analysed the tracks separately for each class and mapped them in Figure 5. In Figure 6, a color coding of windstorm classes is used to present the genesis and lysis locations of the windstorms (the same colors for the classes are used throughout the study). The windstorms arriving from the South-West travel typically the longest distance (Figure 5a,c) which is also consistent with these storms having the longest lifetimes (Figure 4a). In comparison, the tracks of windstorms arriving from North-West (Figure 5b,d) are shorter than the South-Western windstorms (Figure 5a,c), however, in rare cases also NW-windstorms develop in the South Atlantic (Figure 6a). Most of the NW-windstorms develop in the Norwegian Sea, while the SW-windstorms have a more southern genesis location. The storm tracks and genesis locations of class $O_N$ windstorms are a blend of the tracks of SW and NW windstorms, since many of them develop also in the Norwegian Sea but some also much further West in the Atlantic. $O_N$ windstorms (Figure 5h) have typically rather long tracks and a more northern route than the storms in other classes.





**Figure 5.** The tracks of the 92 windstorm cases divided by the class.

The windstorms arriving from South-East and $O_S$ windstorms have a different location of genesis and route of the storm tracks than the ones arriving from Western and Northern directions. Figures 6a and 5e-f show that the SE-windstorms typically develop in Eastern or Central Europe and the storm tracks of SE-windstorms resemble the letter "S", suggesting the presence of a high-pressure system to the north of Finland or Russia. Alternatively, this pattern may indicate that the overall airflow or jet stream is weak, causing the storms to be pushed eastward. The $F_S$-SE and $O_S$ windstorms have similar storm tracks, and the windstorms in these classes typically pass Finland from the Eastern side. However, the development of the $O_S$ windstorms in some cases starts farther South than class $F_S$-SE windstorms (Figure 6a), starting from the Mediterranean Sea and Northern Africa. $O_S$ windstorms have their




lysis locations (Figure 6b) generally farther South than other classes. $F_S$-SE windstorms have short tracks and they
330 tend to have their lysis locations close to Finland.

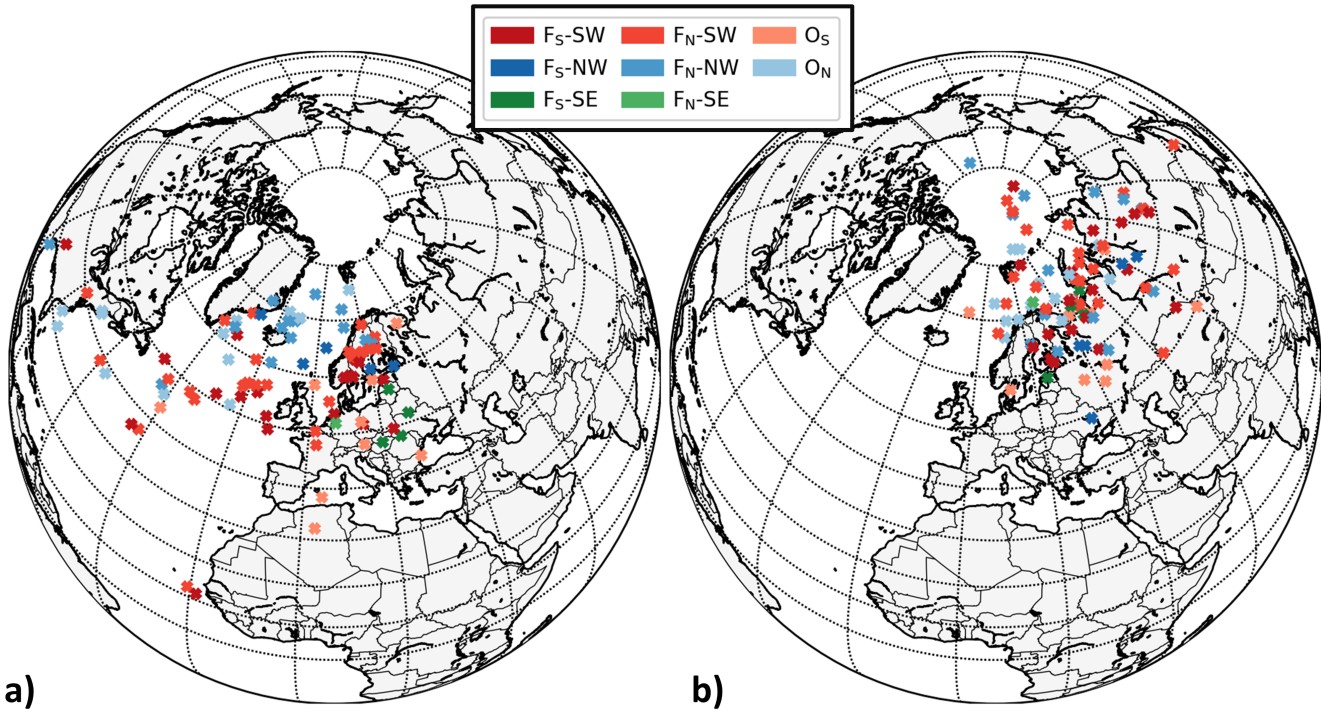

**Figure 6.** a) Genesis and b) lysis of the windstorms in each class.

Many of the $O_N$ storms have their lysis locations to the north of Finland, and over the Barents sea. In general,
for the lysis of the windstorms, there is not a very clear pattern between different classes. The classification is done
based on the arrival direction and thus, possibly for that reason the lysis locations of the storms do not follow as
clearly the class distribution as the genesis location.

335 **4.4  Windstorm impacts on electricity grids**

After analysing the differences in the meteorological characteristics of the windstorms, we now focus on the differences
in the damage on electricity grids. In Section 4.4.1 we first describe the impacts in different windstorm classes. In
Section 4.4.2 we focus on the seasonal impacts and distribution of the windstorms, and finally in Section 4.4.3 the
regional windstorm impacts.





### 4.4.1 Impacts of windstorms and power outage correlation with wind parameters

To quantify the damage of the windstorms in different classes, we calculated the amount of NDP per windstorm case and windstorm type (Figure 7).

From Figure 7 we can conclude that in the analysed period the highest damage per windstorm occurred on average in the NW-windstorm cases: $F_N$-NW and $F_S$-NW windstorms caused on average over 250 000 NDP without electricity. Also, on average, $F_S$-SE windstorms caused high impacts (180 000 NDP/windstorm). The windstorms in the rest of the classes caused 70 000 ($F_N$-SE) to 160 000 ($F_N$-SW) NDP/windstorm. As presented earlier in Sections 4.2 and 4.3, classes differ in their characteristics and their windstorm tracks and genesis/lysis locations. Therefore, also the impacts of the different windstorm class storms have different reasons for their damaging potential. NW-windstorms are the most impactful, likely due to their widespread and strong wind gusts (Figures 4e and f). Furthermore, they have the lowest minimum MSLP (Figure 4b) and they propagate fast (Figure 4c), which indicates also other damaging features such as a gusty and turbulent surface wind layer. $F_S$-SE windstorms also cause high impacts, likely because they affect areas with high population density and dense electrical grids (Southern and South-Western Finland), and they propagate slowly which can accumulate damage and slow down the fixing of the faults. Another possible reason might be easterly winds, as trees are less used to this wind direction, making them more prone to falling. However, the $F_S$-SE class contains only 4 cases which does not allow us to draw clear conclusions. The least damage occurred in class $F_N$-SE and $O_N$ windstorms. In these classes, the wind gusts are not very strong (Figure 4) and they usually do not reach the highly populated areas in Finland.



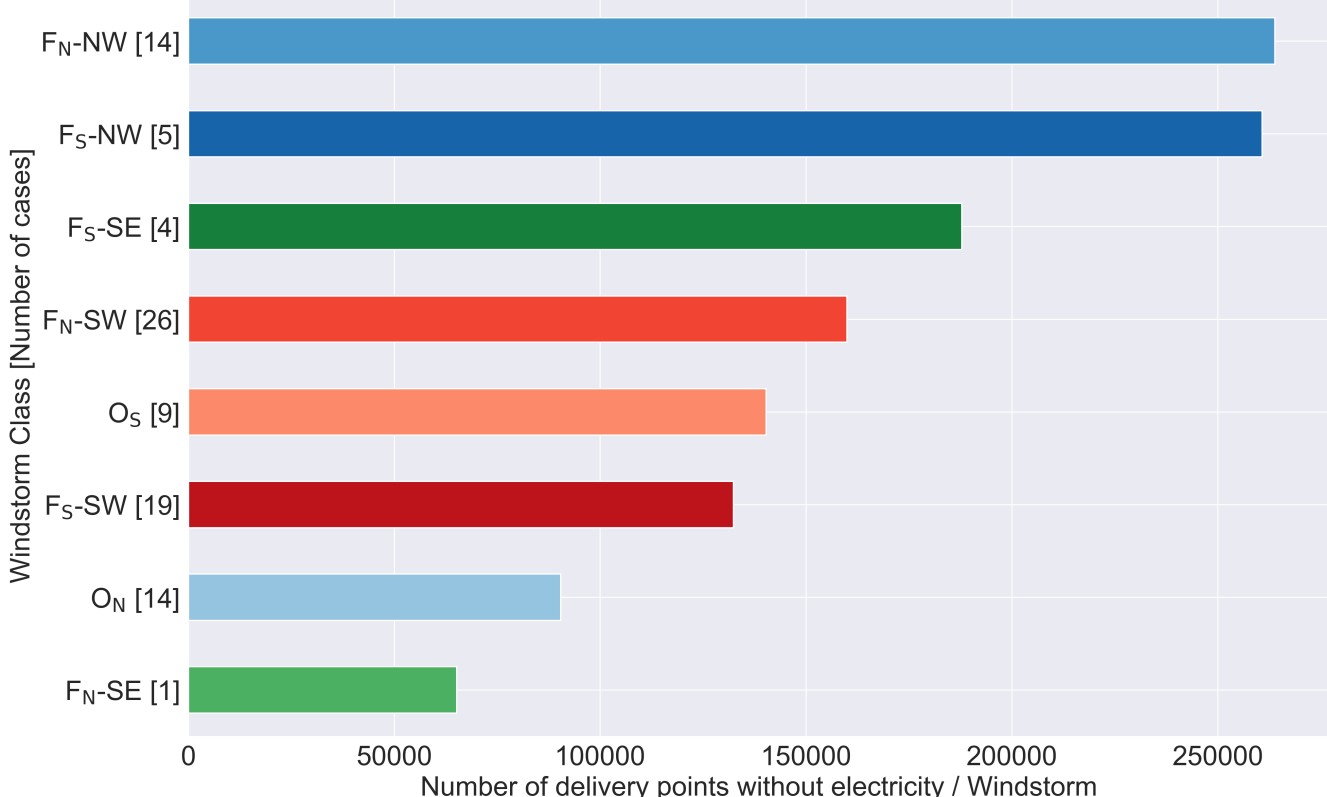

**Figure 7.** Average number of delivery points (NDP) without electricity per windstorm class. The numbers in square brackets on the y-axis represent the number of cases in each class. The colors of the bars represent the colors of the different windstorm class used also in other figures.

To examine the most significant wind-related factors impacting the electrical grid, we computed the NDP and electrical grid fault (hereafter faults) correlation coefficient with ERA5 wind gust parameters and wind gust observations of Finnish Meteorological Institute (Figure 8). The wind parameters included (over the domain of Finland) are: the daily mean of maximum wind gusts at observation stations, the daily mean of maximum wind gusts in ERA5 grid cells, the 90th percentile of daily wind gusts of ERA5, the daily maximum wind gust value in ERA5 and the area in percentage where wind gusts are over 21 m/s. Due to the different characteristics of the meteorological variables, a simple Pearson correlation coefficient was used. The results using the Spearman and Kendall correlation coefficient, which show qualitatively similar results, can be found in the appendix B1. The highest correlation coefficient was found between the area of strong wind gusts and both, NDP (0.61) and faults (0.59), which can be considered to be a strong correlation (Figure 8). The wind gust speed is commonly known to influence the amount of damage. However, in traditional weather forecasting and impact assessment, the spatial extent of strong wind gusts is not





often considered carefully, rather the maximum wind gusts are examined. Based on our results, the size of the area
of strong wind gusts may play a more important role in damage to the electricity network than expected.

Figure 8 shows a medium correlation (0.46) between the observed wind gusts and NDP. Similarly, the 90th
percentile of ERA5 wind gusts reaches a correlation of 0.45 with NDP. The ERA5 maximum wind gust has a
correlation of 0.42 with NDP and 0.39 with faults. The daily mean of maximum wind gusts of ERA5 correlates the
least (weak correlation) of the wind parameters examined with both NDP (0.26) and faults (0.21). It is important
to note also the negative, although rather weak, correlation between minimum MSLP and both NPDs and faults,
indicating that lower MSLP is associated with more damage caused by the windstorm. This is not surprising, as
MSLP is often considered a meteorological indicator of windstorm strength due to its relationship with maximum
wind gusts. All correlations were tested with p-values. At the significance level of 1% (p-value $<0.01$), all parameters
that showed either medium or strong correlations were found to be statistically significant. ERA5 mean of maximum
wind gust and minimum MSLP correlation with both NDP and faults were found statistically significant on level of
5% (p-value $<0.05$). The correlations were not significant (p-value $>0.05$) for parameters with very weak correlation
with NDP and faults (maximum deepening rate and lifetime, for more detail see the correlations and p-values in
Table A1). Finally, the wind gust values of ERA5 and the observed wind gusts were compared in Figure A1 in more
detail, which shows that ERA5 underestimates the most extreme wind gust values compared to the observed wind
gusts.





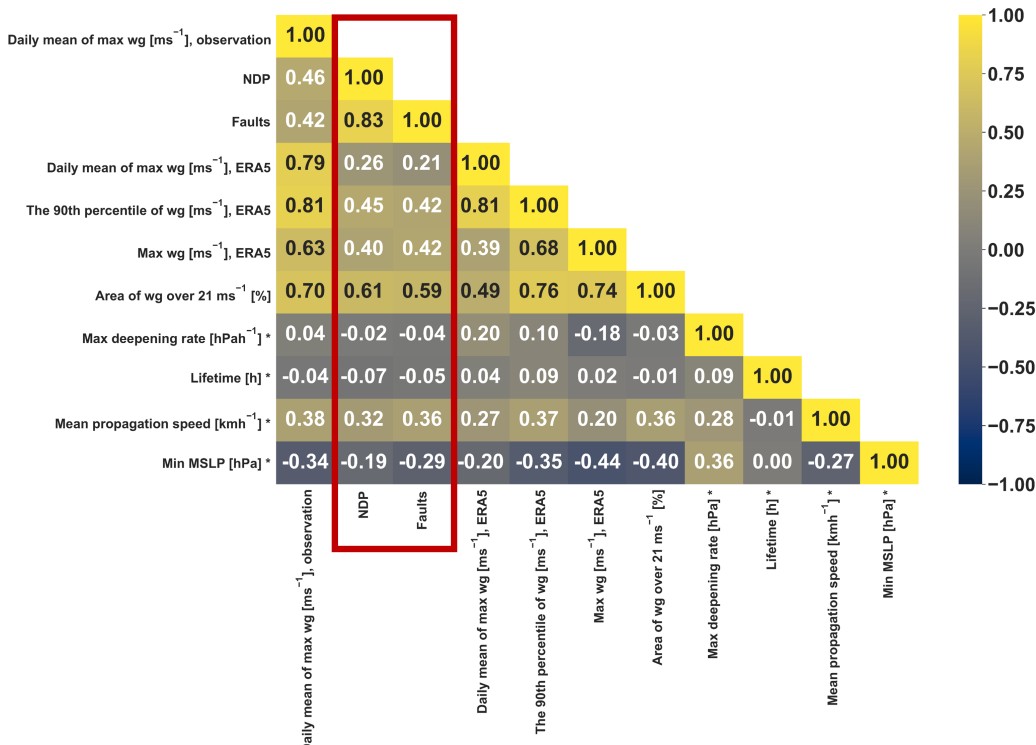

**Figure 8.** Correlation matrix presenting the correlation coefficient values between observed and ERA5 wind gust parameters, windstorm characteristic parameters, NDP, and electrical grid faults. The red box around NDP and fault values highlights the importance of the correlations. The (*) symbol indicates that the correlations were calculated with the characteristics of 69 windstorms ($F_S$ and $F_N$) instead of all 92 windstorms was the case for the other correlations.

### 4.4.2 Seasonality in windstorm occurrence and impacts

To understand better the seasonal impacts of windstorms and the reasons behind the differences, we analyzed the distribution of NDP by season and windstorm class.

In spring (March, April, May) and summer (June, July, August) (Figure 9b,c), the climatological storm tracks
move polewards (Hoskins and Hodges, 2019), and the typical track of an extratropical cyclone is more northerly and they are in general weaker than in autumn (September, October, November) and winter (December, January, February). The difference in impacts between the winter and autumn months may be influenced by factors other than wind speed alone, such as the lack of soil frost, the presence of vegetation canopy, and soil moisture, all of which affect the potential for tree uprooting along power lines.



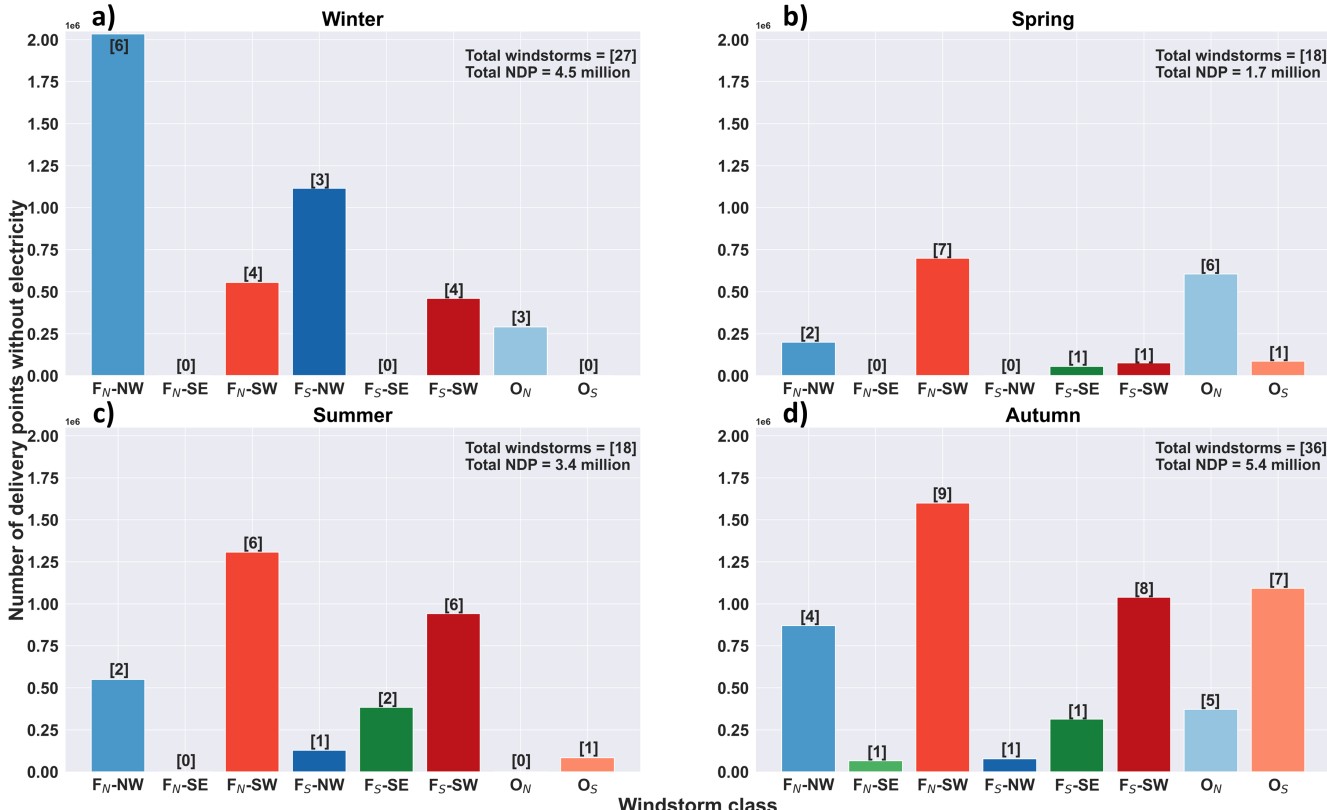

**Figure 9.** Distribution of number of delivery points without electricity (NDP) by windstorm class in a) winter (December, January, February), b) spring (March, April, May), c) summer (June, July, August) and d) autumn (September, October, November). The number of windstorms in each class and season is shown in square brackets.

As Figures 9a and d show, the most windstorm cases occurred in autumn (36) and winter (27). The largest total damage was also related to autumn (5.4M NDP) and winter windstorms (4.5M NDP). This could be due to the fact that, although in winter the windstorms may be more intense than in autumn, the soil frost in winter anchors the tree roots into the ground and prevents the trees from falling. In spring and summer, the number of windstorms is lower (18 windstorms per season), however, during the spring the damage (1.7M NDP) is only half of the damage

in the summer (3.4M NDP). Also this supports the assumption of the significance of soil frost in the prevention of tree uprooting: especially during March (when Finland is still partly covered in snow) the soil frost is often still present and prevents trees from falling in many parts of the country. Additionally, the trees carry leaves in summer which causes the trees to uproot more easily due to a larger area for wind resistance. If we look at the average NDP numbers per windstorm, the most damage actually occurs during the summer (188 000 NDP per windstorm). In

winter the number is 166 000/windstorm, autumn: 150 000/ windstorm and spring: 94 000/windstorm. We conclude, therefore, that even though often windstorms in summer have weaker wind gusts than in autumn and winter, based



on some other environmental factors like vegetation or lack of soil frost, the summer windstorms can be in some cases even more damaging than those that occur in winter.

When looking into the seasonality of different windstorm classes in Figure 9, it seems that in winter NW windstorms are dominant, but there are no $F_S$-SE, $F_N$-SE or $O_S$ windstorms. In all other seasons, also SE and $O_S$ storms are present. If we look into the seasonality of all extratropical cyclones (Figure C1), there are indeed more NW cyclones in winter, although the total number of the extratropical cyclones is not notably larger than in summer. As Figure 6 shows, the genesis of NW windstorms typically start in Northern Atlantic. In winter, windstorms typically develop in the vicinity of the jet stream, and the Northern Atlantic is particularly active with windstorms, which may explain the larger number of NW windstorms and extratropical cyclones. On the other hand, in summer there are no $O_N$ windstorms (and less extratropical cyclones than in other seasons, Figure C1), which might be a result of a more northerly located jet stream during the summer preventing this type of windstorms from reaching Finland. Also, the seasonality analysis of all extratropical cyclones suggest, that SE cyclones are more common in spring and summer than in autumn or winter. Thus, we can conclude that NW windstorms and extratropical cyclones are more 'winter and autumn systems', whereas SE windstorms and extratropical cyclones occur rather in the warm season. The variability of the windstorm (Figure 9) and extratropical cyclone types (Figure C1) types is slightly smaller in winter and autumn than in spring and summer.

Our results support previous studies which have stated that the intensity and extent of the windstorms are stronger during the cold season (October-March) than the warm season (April-September) (Laurila et al., 2021a). The extratropical cyclone activity is typically highest in the North Atlantic during the winter (December-February) and lowest in summer (June-August) (Wickström et al., 2020). Wind speeds in Northern Europe are on average 30% higher in winter than in summer (Laurila et al., 2021b).

### 4.4.3 Impacts by region

Finally, we examined the regional impacts of windstorms. As described in Section 2.2, the power outage data was provided for 5 areas in Finland. The areas have different features that also influence the potential for impacts. Area 1 is the most densely populated, including for instance the urban areas of the capital Helsinki and Turku. The process of placing cables underground is most advanced in such urban centers. Areas 2 and 4 include a few bigger cities as well as rural areas. In area 3, there are a few cities, but a large share of the electricity grid is above the ground and often located in highly forested areas. Area 5 is the least densely populated and less forested than any of the other areas.



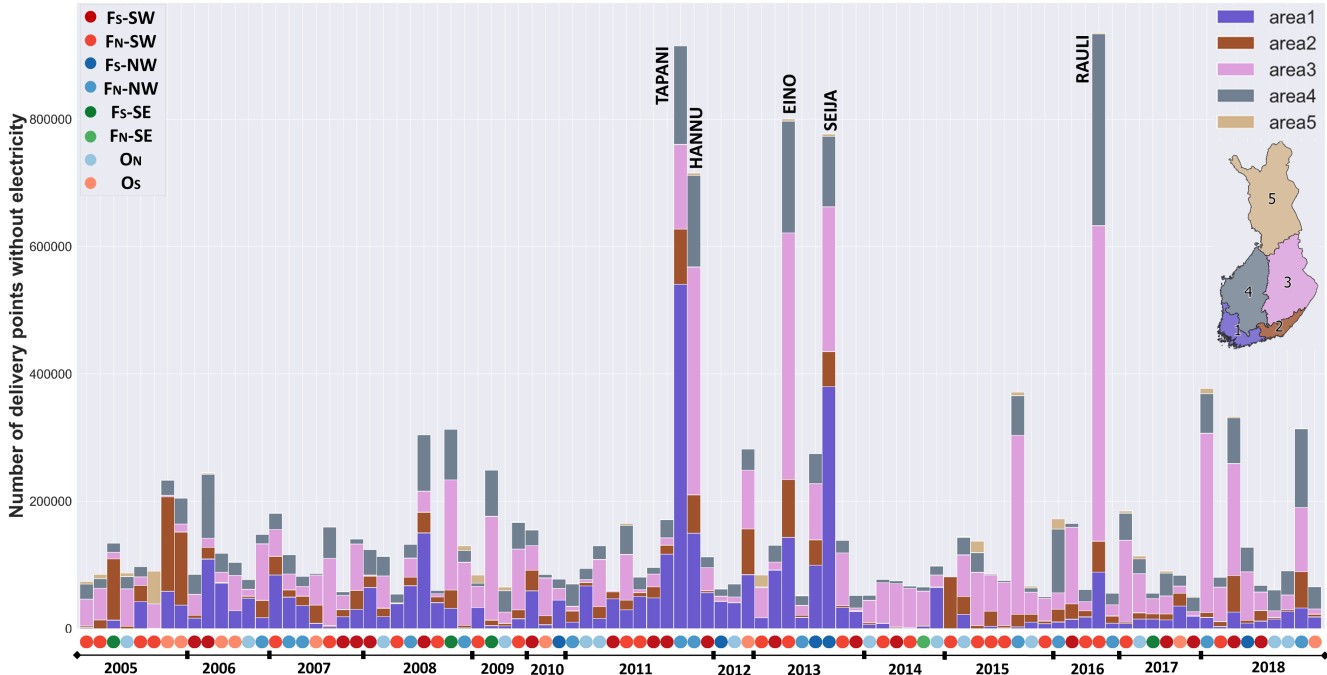

**Figure 10.** Numbers of delivery points for each of the 92 windstorm cases. The colored circles below the bars indicate the windstorm class. The Finnish names of the five most damaging windstorms are shown on top of the corresponding bars.

Most of the damage occurred in Eastern Finland (area 3), most likely because of the structure of the grid (many overhead lines) and the highly forested terrain. As Figure 10 shows, until 2013 a lot of damage occurred in South-Western Finland (area 1), after which the NDP number in area 1 distinctly decreased. This may be partly explained by the development in the ground cabling especially in area 1 (Jasiūnas et al., 2023b). Ground cabling is becoming more common gradually in other areas as well, but due to the challenging terrain and cost of ground cabling, it is not profitable (or even possible) to transfer the entire grid below the ground, especially in more rural areas (areas 3, 4 and 5). The windstorms that cause damage also in area 5 are in general either $F_N$ or $O_N$ storms, since only in these classes, the strong wind gusts reach this far north. In extreme cases, like the five strongest windstorms labelled in Figure 10, the wind gusts are usually widely spread and therefore cause damage over large areas. However, also in these cases there are differences depending on the route of the windstorm.

$O_S$ windstorms cause damage mostly in areas 1, 2, and 3. As Figure 5g shows, the route of $O_S$ windstorms is typically located on the Southern side of Finland, and since the strongest wind gusts primarily remain on the southern side of the track therefore only a small part of the wind gusts reach southern and eastern parts of Finland. In the case of $F_S$-SE windstorms the situation is similar and they affect mainly areas 2 and 3 (in a few cases also areas 1 and 4). $F_S$-SE windstorms's arrival direction is similar to $O_S$, and the wind direction varies from southern to eastern and impact areas are similar. The location of the arrival plays a role in the regional distribution of damage.





For instance, Southern Finland ($F_S$) windstorms almost never cause damage in area 5. Although Windstorm Seija (class $F_S$-NW) was a very intense windstorm, but it caused nearly no damage in area 5 (Figure 10).

We can conclude that the route of the windstorms plays a role, however, the most important thing in the damaging power of the storm is the overall strength of the windstorm including the wind gust speed. In addition, the season and environmental and climatological factors have an effect on the resulting damage, e.g. the top 5 most damaging windstorms have occurred when there was no soil frost present to anchor the trees to the ground.

## 5    Conclusions

This study examines windstorms and extratropical cyclones that impacted Finland from 2005 to 2018, focusing on
their meteorological characteristics, effects on the electrical grid, and seasonal and regional variations.

We seek to develop a focused understanding of windstorm impacts that goes beyond conventional meteorological classifications. The goal is to improve preparedness for future windstorm risks, offer aspects for strengthening grid resilience, and provide tools for assessing and forecasting windstorm impacts in Finland and comparable regions. To address these objectives, we created a novel classification method for extratropical cyclones, focusing on their
arrival location and direction. We further selected windstorms based solely on their impact, specifically power outages, rather than relying on meteorological parameters. If impact data is available, this classification can be applied beyond Finland, making it relevant for other regions affected by windstorms as well. We then analyzed the meteorological characteristics of windstorms in comparison with extratropical cyclones by class, studied the windstorm-related meteorological properties, and assessed how impacts vary based on the type of windstorm and its
meteorological attributes. We examined how the impacts differ by region and season across different extratropical cyclone classes.

Our findings show that the most common type of extratropical cyclone that affects Finland originates from the South-West (classes $F_N$-SW and $F_S$-SW, 55%). Notably, class $O_N$ and $O_S$ windstorms, which account together for 25% of all damaging windstorms, illustrate that damage can occur even if the windstorm's low center does not
directly move over the country. Although $F_S$-SE windstorms are relatively rare (4%), their impact is significant when they do occur.

The characteristics and impacts of these extratropical cyclones are closely linked to their origin and movement. Windstorms from the southwest (classes $F_N$-SW and $F_S$-SW) travel longer distances and cause in total the most damage. Some of these type of windstorms may have originally been tropical cyclones that transitioned to extra-
tropical cyclones, as indicated by their storm tracks. Extratropical cyclones arriving from the northwest (classes $F_N$-NW and $F_S$-NW) mainly originate in the Norwegian Sea, while $F_S$-SE and OS windstorms come from Eastern and Southern Europe. $F_N$-NW and $F_N$-SW windstorms create severe damage due to their higher wind gust speed. $F_S$-SE storms, being slower propagating, tend to cause accumulating damage over time, whereas $F_N$-NW and $F_S$-NW windstorms, in particular, are characterized by their extreme strength regarding maximum wind gust,



minimum MSLP and widespread gusts in highly populated areas like Southern Finland. In the analyzed period, class $F_N$-NW and $F_S$-NW-windstorms caused the highest average number of delivery points (NDP) without electricity. $F_S$-SE windstorms also caused significant impacts, particularly in densely populated areas with extensive electrical grids. Wind gusts and the spatial extent of strong winds were identified as major contributors to grid damage, with larger areas of strong gusts showing higher correlations with electrical grid key parameters: NDP and faults. The

results emphasize the importance of considering both wind speed and the area affected by strong gusts in impact assessments.

    Seasonal patterns indicate that windstorms are more frequent and the total damage is higher in autumn and winter compared to spring and summer. The variability of the damaging extratropical cyclones is higher in spring, summer, and autumn, reflecting a broader range of cyclone classes in these seasons than in winter. In autumn and

winter, while windstorms can be intense, factors like soil frost reduce damage by preventing tree uprooting, whereas summer windstorms, though typically weaker, can cause significant damage due to foliage and lack of soil frost. This highlights the importance of considering environmental conditions beyond wind speed when assessing windstorm impacts.

    Regional analysis shows variability in windstorm impacts across Finland. Areas with overhead power lines and

dense forests, particularly Eastern Finland, experience more frequent and severe damage. A shift in damage patterns over time may also be influenced by increased ground cabling in urban areas. The route and strength of windstorms also affect regional damage distribution, with certain storm classes causing concentrated impacts in specific areas. With climate change potentially shifting the jet stream northward (Bengtsson et al., 2006; Haarsma et al., 2013), Finland might experience an increase in powerful windstorms from northwesterly and southwesterly directions,

southwesterly windstorms being currently the most common. Climate change will also likely impact the soil frost conditions in the future (Kellomäki et al., 2010; Gregow et al., 2011; Lehtonen et al., 2019). Recent studies suggest a negative correlation between soil frost and wind-related power outages (Haakana et al., 2024; Láng-Ritter et al., 2023) i.e. soil frost prevents the uprooting of trees and reduces the number of power outages. Future research should study these insights in more detail and climate change aspects to refine impact assessments and improve forecasting

and mitigation strategies. Other environmental factors, such as soil moisture and vegetation, also play a role in windstorm impacts, highlighting the need for comprehensive, multivariate analysis.

    Using impact data is crucial for understanding windstorm damage, developing new impact models, and enhancing meteorological tools. Challenges such as data reliability, variability, and limitations of current datasets need to be however addressed (Láng-Ritter and Mäkelä, 2021). In summary, this study improves our understanding of windstorm

risk and extratropical cyclone impacts, highlighting the importance of seasonality and regional characteristics. Future research should explore the relationship between meteorological conditions, and societal, and environmental factors to enhance resilience against windstorm impacts. Extending the work to include analysis of vulnerability (e.g. electricity grid located close to trees), and exposure (the part of the electricity grid above the ground) would further improve the understanding of windstorm risks.



*Data availability.* The extratropical cyclone tracks in Northern Europe for 2005-2018 are available on Zenodo:

https://doi.org/10.5281/zenodo.13807849.

Copernicus Climate Data Store for downloading the ERA5 data:

https://doi.org/10.24381/cds.bd0915c6

The wind observation data of FMI can be downloaded using the open data portal of FMI:

https://en.ilmatieteenlaitos.fi/download-observations.

The authors do not have property rights to distribute the power outage data other than in scientific cooperation.

## Appendix A

### A1

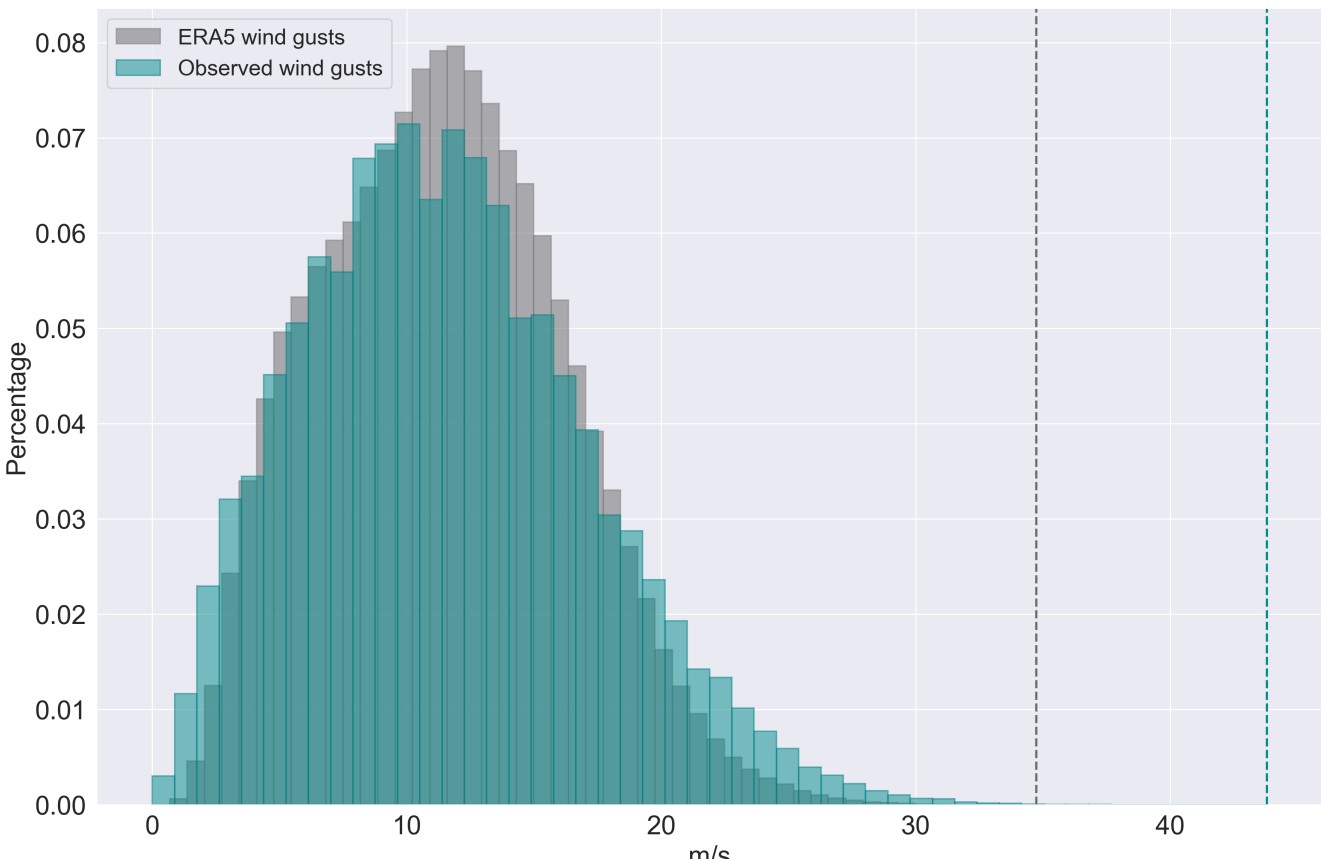

**Figure A1.** Comparison between ERA5 reanalysis and observed wind gust values in Finland, 2005-2018. ERA5 wind gusts are represented with grey and observed wind gusts with green color. The dashed lines represent the maximum wind gust speed value for both datasets. The ERA5 wind gust values represent all recorded wind gusts within the Finland box during the windstorm dates.



| Correlated parameters | Correlation | p-value | Sample size |
|---|---|---|---|
| ERA5 mean of 90th perc wg, NDP | 0.450 | 6.94e-06 | 92 |
| ERA5 mean of 90th perc wg, faults | 0.417 | 3.54e-05 | 92 |
| ERA5 wg area, NDP | 0.609 | 1.22e-10 | 92 |
| ERA5 wg area, faults | 0.589 | 9.02e-10 | 92 |
| ERA5 wg mean of max, NDP | 0.260 | 0.012 | 92 |
| ERA5 wg mean of max, faults | 0.207 | 0.048 | 92 |
| ERA5 max wg, NDP | 0.399 | 8.40e-05 | 92 |
| ERA5 max wg, faults | 0.424 | 2.55e-05 | 92 |
| Obs mean of max wg, NDP | 0.460 | 3.89e-06 | 92 |
| Obs mean of max wg, faults | 0.412 | 3.20e-05 | 92 |
| Max deepening rate, NDP | -0.018 | 0.885 | 69 |
| Max deepening rate, faults | -0.04 | 0.760 | 69 |
| Lifetime, NDP | -0.07 | 0.555 | 69 |
| Lifetime, faults | -0.05 | 0.671 | 69 |
| Mean propagation speed, NDP | 0.317 | 0.008 | 69 |
| Mean propagation speed, faults | 0.361 | 0.002 | 69 |
| Min MSLP, NDP | -0.186 | 0.127 | 69 |
| Min MSLP, faults | -0.289 | 0.016 | 69 |

**Table A1.** Pearson's correlation coefficient for different parameters and p-values.



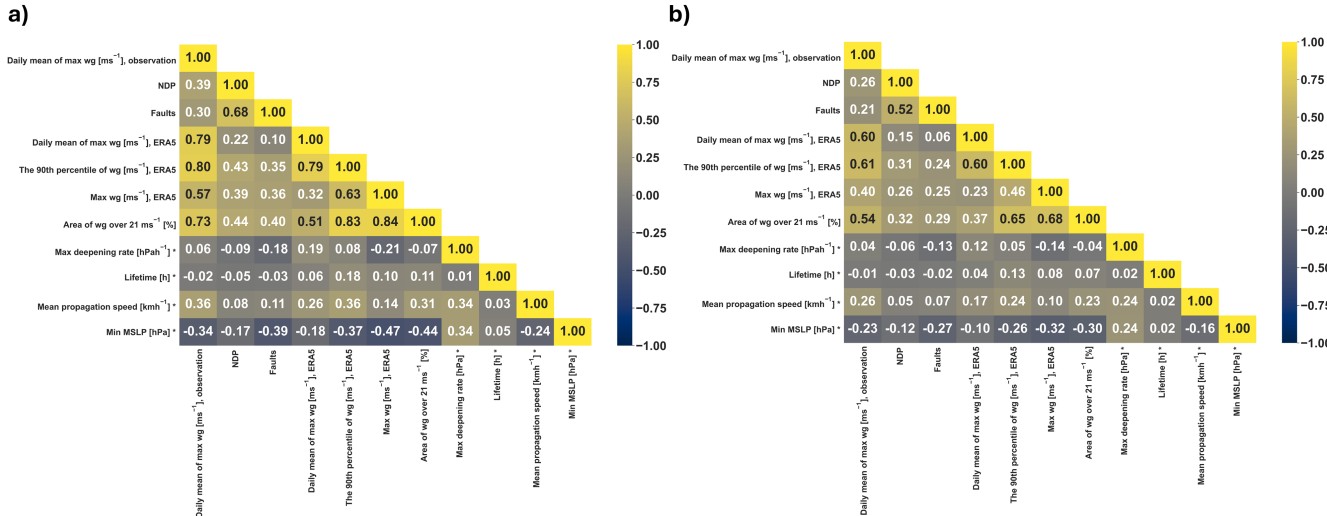

**Figure B1.** a) Spearman and b) Kendall correlation coefficient values for meteorological parameters.

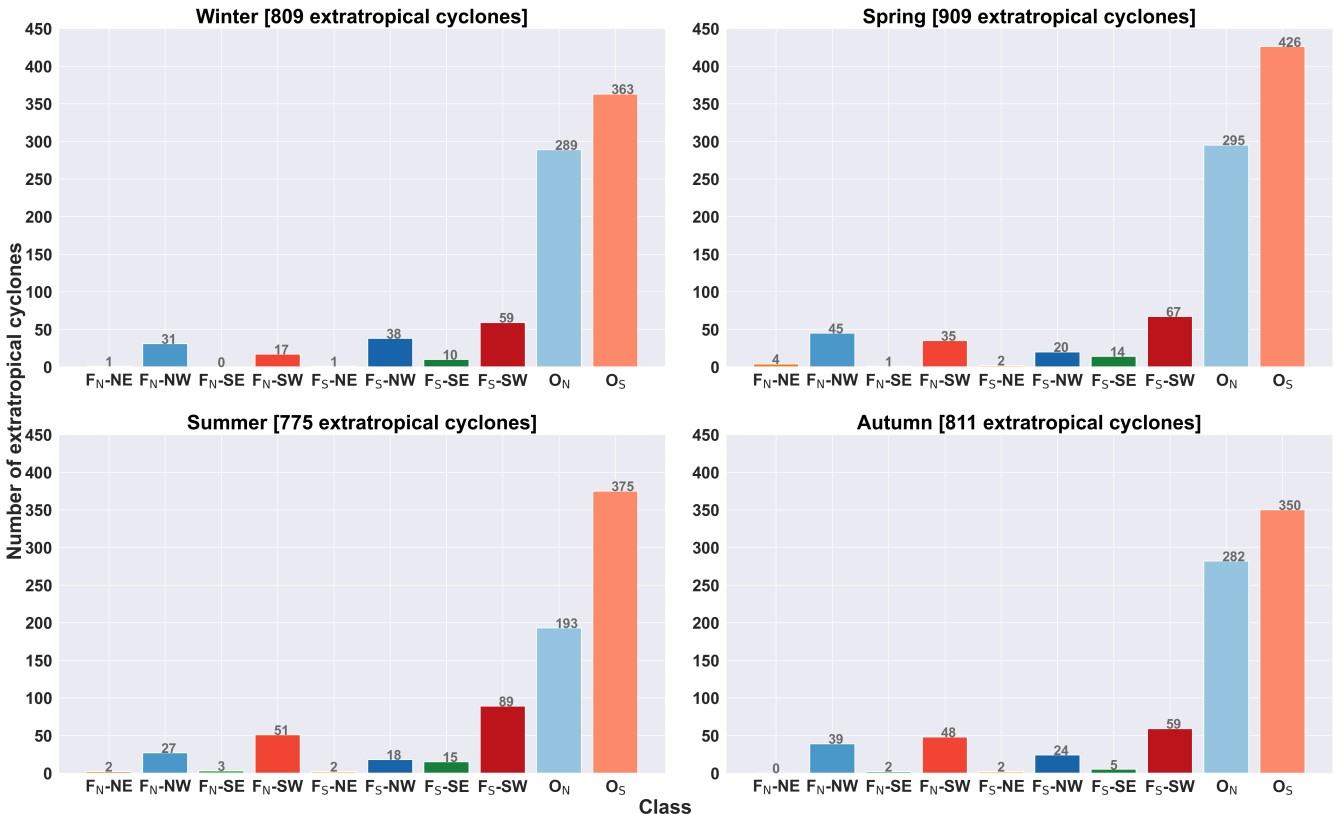

**Figure C1.** The seasonal distribution of all extratropical cyclones during 2005-2018. The number inside the brackets in the titles represents the total number of extratropical cyclones during the season.

*Author contributions.* ILR conceptualized the classification method, filtered and analyzed the power outage data, and wrote a significant part of the article. TKL tracked the extratropical cyclones, extracted their characteristics, contributed to writing the meteorological sections, and revised the article. AM and HG gave advice, revised the article, and provided funding. VAS provided supervision, advice and guidance on scientific viewpoints and edited and revised the article.

*Competing interests.*

The authors declare that they have no conflict of interest.

*Acknowledgements.* Research for this paper at the Finnish Meteorological Institute was funded by the European Union's Horizon Europe Programme (project CREXDATA, grant agreement number 101092749), NordForsk (project NordicLink,



project number 98335), and the LODE project (2018-2021, European Commission - DG ECHO). The research at the University of Helsinki was supported by the Academy of Finland (grant no. 338615)2. The authors would like to thank Enease Oy for providing the power interruption data and insights into the electricity distribution system in Finland. Additionally, the authors

thank sincerely Dr. Josias Láng-Ritter from Aalto University for his valuable comments and assistance in editing the article. The language of the paper was in some parts revised using ChatGPT, however, ILR reviewed and edited the suggested texts and takes responsibility for this publication's final content.





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
