# Peer review of "Classifying extratropical cyclones and their impact on Finland's electricity grid: Insights from 92 damaging windstorms"

_EGUsphere, 2024_

## Author Comment (AC1)

**Reply for 1# Referee:**

The paper investigates the impact of windstorms on Finland's electricity grid. The authors present a new classification method for identifying windstorms, which is based on the impacts (here: power outages) rather than on meteorological aspects. They analyze the different windstorm classes in terms of their meteorological properties and impacts, focusing on regional and seasonal differences.

The paper covers an interesting and relevant topic. It is well written and clearly structured. Apart from some comments below, I can recommend the paper for publication and I feel it will provide a useful contribution to the field.

We thank the reviewer for their constructive comments and are pleased to hear that they found the manuscript interesting. We have copied all detailed comments below in blue and provided our responses to each specific comment in black.

1. When I read the paper, I was quickly confused about how you differentiate between extratropical cyclones and windstorms. Only when I got to Section 3.2 on page 7, I was sure that windstorms are those extratropical cyclones that cause high power outages. Maybe it would be good to define this as early as possible.

   Reply: In the Introduction, we have clarified this on lines 68-70 as follows: *"This study examines the impacts of extratropical cyclones and windstorms on Finland's electricity grids between 2005 and 2018. Out of the 3,304 extratropical cyclones examined, 92 are considered significant windstorms by having caused at least 50,000 power outages (Section 3.2)."*

2. Section 3.2

   1. How many cyclones did you analyze in total - in addition to the 92 windstorms?

      Reply: In lines 166-167 (section 3.1) of the original manuscript we state that after we apply the different filters to all of the objectively determined cyclone tracks, we have 3304 extratropical cyclones. This is the total number of cyclones that we analysed. We have revised this sentence to make this clearer.

   2. Line 187: You state that you exclude dates if the same windstorm caused power outages during consecutive days. Which dates did you exclude in these cases - the whole storm or just one of the days? And which day did you keep? Please clarify.

      Reply: We excluded "the less significant date" and included the day which had the largest impacts i.e. the largest number of delivery points without electricity. In lines 200-201, we added an additional sentence to clarify this: *"In the case of (1), we retain the date that had the largest NDPs."*

We included for instance Tapani and Hannu windstorms although they occurred on two consecutive days, because there were clearly two separate storm tracks and cyclones identified.

3. Section 3.3: What are the borders of the "Outside Finland" domain?

Reply: The borders of the class O windstorms are the domain of extratropical cyclones potentially affecting Finland (0–60°E, 50–75°N.)". This is mentioned in section 3.1 on the lines 177: "...the tracks that passed through the domain of 0–60°E, 50–75°N."

For improving clarity, we have added the same information into section 3.3, where we explain the classification. You can find the additional information in the caption of Figure 2 and in lines 209-210: "...whether it does not pass directly across Finland at all (class O, Outside of Finland, but inside of 0–60°E, 50–75°N box)."

4. Figures 3 and 4: Would it make sense to use the same color coding for windstorms and extratropical cyclones in both figures?

Reply: We appreciate the comment and have now unified the color coding of the two figures. Figure 3 has been updated.

5. Section 4.4.1: How exactly did you compute the correlation coefficient between ERA5 and NDP or between observations and NDP given their different spatial scales?

Reply: We used the daily sum of NDP and faults to calculate the correlation coefficient with the selected meteorological parameters. For example, we calculated the mean of the maximum wind gusts across all FMI's observation stations in Finland to maximize spatial coverage for each windstorm day and correlated it with the daily NDP sum. A similar analysis was done with ERA5 data, using the mean of the maximum wind gusts across all grid cells within the Finland domain (20.0–31.4°E, 59.5–70.1°N). Likewise, correlations with NDP and faults were calculated for other meteorological parameters, such as minimum MSLP and propagation speed.

We have revised the sentence in lines 367-374 to make it more clear: "To examine the most significant windstorm-related factors impacting the electrical grid, we computed the correlation coefficients between the daily sums of NDP and electrical grid faults with ERA5 wind gust parameters, wind gust observations from the Finnish Meteorological Institute (Figure 8), and daily storm track parameters (e.g., minimum MSLP) over the Finland domain (20.0–31.4°E, 59.5–70.1°N). For example, we calculated the mean maximum wind gusts across all FMI observation stations for each windstorm day and correlated them with the daily NDP sum. A similar analysis was performed using ERA5 data by computing the mean maximum wind gusts across all grid cells within the Finland domain. Similarly, correlations were also computed for other parameters, such as minimum MSLP and propagation speed."

Section 5

1. Could your classification method be applied to any other country in Europe given that the impact data is available? Or would one need a comparable country size, cyclone statistics, ...?

   Reply: The classification method could likely be applied especially to other northern European forested countries with available impact data, though some adjustments may be needed. For instance, if applied to a smaller country than Finland, the initial classification step dividing the country into two regions could be omitted. In Mediterranean regions, modifications may also be required, for example, include tracking shorter-duration windstorms, such as Mediterranean cyclones, which often have shorter storm tracks.

   We have included a sentence in the end of Conclusions, lines 545-546: "*The classification method could likely be applied especially to other Northern European forested countries, though some adjustments may be needed.*"

2. Could your method be used to estimate the impacts of windstorms on power outages under climate change conditions?

   Reply: Possibly yes. For example, our methodology could be applied to climate model output, although first a careful examination of how well windstorms are resolved by the climate model would be needed. As this work is beyond the scope of this current study, and somewhat speculative, we have not added any text to the manuscript on this topic.

---

## Author Comment (AC2)

**Reply for 2# Referee:**

The study of Láng-Ritter et al. investigates extra-tropical cyclones and their impact on the electricity grid in Finland between 2005 and 2018. The authors analyse 92 selected cyclones, which they define as windstorms, where power outages in Finland can be registered.

The study compares a set of cyclones with windstorms (impact in Finland) by means of selected cyclone criteria. The authors use a classification approach where the location and propagation direction of the cyclone into the target region is used.

The manuscript is well written and structured. The reader can nicely follow.

I have only one bigger comment: in my opinion the authors slightly exaggerate the value of their study. The manuscript gives insights in parts of characteristics of extra-tropical cyclones and windstorms over Finland which is definitely interesting and relevant. The authors write in the conclusion that the aim of the study is to improve preparedness for future windstorm risk, provide tools for forecasting windstorm impacts, etc.
There are definitely further steps to do to achieve these goals.
I suggest to clarify it in the conclusion. (see also my last minor comment)

Reply: We thank the reviewer for their constructive comments and are pleased to hear that they found the manuscript well written, structured and interesting. We have copied all detailed comments below in blue and provided our responses to each specific comment in black.

We agree with the general comment regarding the preparation for windstorm risks and have moderated our conclusions and other parts of the manuscript accordingly, acknowledging in hindsight that they slightly exaggerated what our study had achieved. Based on this and your last comment, we have now revised the last paragraph of Conclusions as follows, line 532-545:

"*Using impact data is crucial for understanding windstorm damage, developing new impact models, and enhancing meteorological tools. However, challenges such as data reliability, variability, and limitations of current datasets need to be addressed (Láng-Ritter and Mäkelä, 2021). Although our results are not directly usable for operational impact forecasts or/warnings, they demonstrate the high potential of such new meteorological applications. An operational impact model (likely machine learning –based) would need to be trained with an extensive dataset of meteorological, environmental and impact variables, and it would need to be tested and validated in real cases. In a national Finnish context, this would be possible to establish due to the availability of impact (i.e., outage) data, but unfortunately for larger regions such a service would be difficult to establish due to limitations in the impact data availability.*

*In summary, this study improves our understanding of extratropical cyclones and windstorm impacts, highlighting the importance of seasonality and regional characteristics. Future research should explore the relationship between meteorological conditions, and societal, and environmental factors to enhance resilience against windstorm impacts. Extending the work to include analysis of vulnerability (e.g. electricity grid located close to trees) and exposure (the part of the electricity grid above the ground) would further improve the understanding of windstorm risks."*

Additionally some minor changes on lines:

L71-72: The previous manuscript mentioned windstorm risks, but we have now toned this down since we are not dealing directly with risks.

"*The goal is to enhance preparedness for future windstorms and wind risks...*"

to

"*The goal is to enhance preparedness for future windstorms...*"

Please find below further minor comments which I recommend to consider before publication.

L 5: The impact is purely on power outage, correct? Can you include it to avoid misunderstanding.

Reply: Yes, that is correct. We have now revised the abstract accordingly and the text in line 5 now reads "... *we select them based on their impacts, namely power outages, to reach a more....*"

L. 45: MSLP is typically used for mean sea level pressure.

Reply: Thank you for noticing this, we have changed the definition of MSLP on this line to mean sea level pressure.

L. 98: The 3 s wind gust is computed every time step. Isn't this the internal model time step instead of the time resolution of the data, which is 1h?

Reply: This was an issue of terminology – there was some confusion between the model internal timestep and the output frequency. This sentence has been revised (lines 100-102) and now reads "*The 3 s wind gust is computed at every model timestep (12 minutes) and the maximum value since the last post-processing period is output. In ERA5, the post-processing period is 1 hour and thus we obtain the maximum 3 s wind gust that occurred in previous 1 hour.*"

L. 139: is it possible to see a clear relationship with NDP and wind storms? Can it happen that one failure which is related with a wind storm leads to a different numbers of NDP dependent on the location of the failure within the electricity grid. Are there more and less vulnerable locations in the grid? This would include further random effects in the relationship.

Reply: You are correct, it is exactly like this. The vulnerability of the Finnish grid to windstorms varies significantly by region. This variation is influenced by the grid type (overhead vs. underground cabling), environmental factors, and maintenance practices (e.g., proximity to trees and tree species). For NDP, population or household density also plays a role.

In an operational impact forecasting model, network structure and other vulnerability factors should be included, ensuring that observed impacts at specific locations align with corresponding environmental and meteorological data. However, in our study, impact data is compiled for larger regions due to the unavailability of more precise data, making this approach unfeasible.

L. 186: is your comparison of days with failure and the occurrence of windstorms fully subjective? When do you decide that there is no cyclone around (case 2), line 187) and when is a cyclone near enough to potentially have impact on the outage?  Are there objective (transparent) criteria to decide? Can you convince the reader that this validation works well, e.g. by adding meaningful examples to the supplement?

Reply: We decide that there is no cyclone present if no objectively identified cyclone track is detected in the area 0–60°E, 50–75°N. By visually inspecting synoptic analysis charts, this approach (and the size of the box) was found to be valid in almost all situations. In a very small number of cases, there were days with >50,000 NDP but no cyclone objectively identified by the TRACK software in this box. This absence is most likely due to limitations of the TRACK algorithm, particularly when the cyclone is very stationary or short-lived (i.e., moving less than 500 km during its lifetime).

Our method can be considered mainly objective because we use the observed outages as the filter; if indeed > 50,000 NDPs are recorded labelled in the data as "due to the wind", we can safely assume there must have been a windstorm.  On the other hand, it is true that the method is subjective because we set NDP > 50,000 as a threshold—why not 10,000 or 60,000? However, if we must define a limit, we had to do so somehow, and this limit was chosen based on expert comments. In our study we additionally assume that the outage data is correctly labelled, which might sometimes not be the case (i.e., they are not due to a windstorm but to convective storm).

We added some extra details to the line 198-199 (case 2): "no extratropical cyclone tracks were found within the defined area (0-60E, 50-75N)."

L. 196: how is the cyclone classified if it passes both boxes F_N and F_S? This would be exactly the blue example in Fig. 2-2.

Reply: The classification is based on the first-entered box. We have included a clarification to the Fig. 2 caption and to lines 212-213: *"In case the extratropical cyclone passes over Finland, it is classified to class FN (Northern Finland, dark blue box) or class FS (Southern Finland, red box) according to which box it entered first."*

L. 236: how is it possible that strong winds do not effect Finland but the cyclone leads to an outage?

Reply: This sentence "*for OS windstorms, the strong winds often remain outside of the domain of Finland,* and when they do extend over Finland, they rather impact Southern or South-Eastern Finland" was possibly mis-leading, particularly the part in italics. Our intended meaning was that for OS windstorms, a large share of wind gusts remain outside Finland because the strongest wind gusts often occur on the southern side of the low-pressure center. However, the OS windstorms included in our list have caused significant impacts (>50,000 NDP) in Finland, typically in Southern and South-Eastern regions.

We have now clarified it on lines 249-251 : *"In contrast, for OS windstorms, a large share of wind gusts remain outside Finland (strongest wind gusts often occur on the southern side of the low-pressure center)."*

L. 260: F_N-NW has the shortest lifetime and F_N-SW the longest. How can this be understood since the difference of the classes is only the more northerly or southerly propagation? Probably the cyclones originate in different locations.

Reply: That is how we understand it as well and this can be observed in Figures 5c and 5d. F_N-SW windstorms tend to originate farther south and west, whereas F_N-NW windstorms typically form near Iceland.

Fig. 4: you are not using all classes out of theoretically 8 classes. You should mention that. I am wondering how reliable the box plot is, e.g. for F_S-NW and F_S-SE including 5 and 4 cases, respectively.

Reply: We have now mentioned this on lines 271-274: *"In the characteristic comparison, we only considered classes F because a comparison with classes O would not be fair. This is due to the large size of the bounding box including numerous cyclone tracks in class O that do not affect the domain of Finland at all. Also, classes FN_NE and FS_NE are omitted here because they did not contain any windstorms."*

L. 315: Fig 6a and b show one case each which has its origin in a very southerly position (around 10°N). This is still the North Atlantic, isn't it? Please reformulate.

Reply: You are correct, the North Atlantic refers to that part of Atlantic > 0°N. We have reformulated as follows:"...*however, in rare cases also NW-windstorms develop in the southernmost regions of the North Atlantic (Figure 6a).*"

L. 353: you have not analysed the wind direction but the propagation of the cyclone. It has not to be the same. It is interesting that both SE classes (F_N, F_S) show a huge difference in NDP. Do you have an idea. Probably your statement is true, that the classes contain 4 and 1 events which is much too small in order to draw clear conclusions.

Reply: Regarding the winds, you are correct that we have not analyzed wind directions. However, by visually inspecting synoptic weather charts we see that windstorms associated with an south/easterly flow tend to have more easterly winds than those arriving from the western sector. Additionally, we also think given the relatively small number of windstorms in both of the classes, we should not overly interpret these results, however, if we were to speculate, we think that population density may also influence the results in different areas of Finland. For example, Figure 5f shows that the F_N windstorm followed an unusual route and mainly affected the eastern part of the country (see Figure 10), whereas F_S windstorms impacted the southern part, which is more densely populated than eastern Finland.

We have reformulated, lines 369-370: "...*possible reason might be the easterly direction of arrival with more easterly winds, as trees are less used to this direction, making them more prone to falling.*"

L. 379: can you exactly define the threshold for medium and strong correlation?

Reply: The definition and interpretations for correlations can vary slightly depending on the field of study, however, the general guideline is that the strength of a correlation is classified as weak (0 to 0.3), moderate (0.3 to 0.5), or strong (above 0.5). More specific guidelines exist, but the interpretation depends on the context and what is being measured. We modified the text on lines 386-387 and have included reference to clarify: *"Correlation definitions vary by field of study, but generally, they are classified as weak (0 to 0.3), moderate (0.3 to 0.5), or strong (above 0.5), which we also use in this study (Turney, 2022)."*

L. 396: I agree the argument that soil condition can influence tree fall. But Fig. 9 agrees with the general relationship between number of storms and number of NDP (largest number of storms and NDP in autumn, second largest of both in winter).

Reply: This is a good point, and we have now added additional analysis and discussion to the text to support our claim:

*Lines 419-427: "Among the windstorms analyzed in this study, autumn storms had similar median in minimum MSLP of 975 hPa and winter storms 977 hPa, however, winter storms produced slightly stronger median wind gusts 28.3 ms-1 compared to 26.8 ms-1 for autumn*

*(analysis not shown). This suggests that soil frost in winter anchors tree roots more effectively, preventing trees from falling, as recently demonstrated by Haakana et al. (2024)... ...Additionally, summer windstorms show slightly higher median in minimum MSLP and maximum wind gusts than spring windstorms, with values of 988 hPa and 23.6 ms-1 compared to 980 hPa and 23.0 ms-1, respectively."*

L. 421: doubling of word "types"

Reply: Thank you, we have corrected this.

L. 462: The goal you are highlighting here, is a nice motivation for upcoming studies. But you are not addressing it with your current study. Your classification approach allows to distinguish between cyclone characteristics of windstorms (impact cyclones in your definition) in the different regions and propagation directions and cyclones (without impact). This is a nice way to learn about these cyclone properties. To learn about risks and to perform impact forecast, you would need an impact model. This can be more or less complex but needs at least a description of a relationship between meteorological parameters (e.g. gust, soil temperature, precipitation, ...) and impact (NDP in your case). You are not analysing when a characteristic leads to impact, when not, and when characteristic leads to no impact (false alarm). This would go into the direction of understanding risk. I agree, further research is needed to achieve this aims.

Reply: Thank you, we do agree with your comment and it has interesting aspects. Furthermore, we have revised lines 489-491 as follows: *"The goal is to improve understanding of the relationship between windstorm characteristics and impacts and to propose new methods for distinguishing impacting windstorms from non-impacting ones, as well as for assessing windstorm impacts in Finland and similar regions."*

---

## Author Comment (AC3)

**Reply for 3# Referee:**

The manuscript presents an interesting an thorough analysis of the impacts of windstorms on Finland's electricity sector. Although in principle arbitrary, their classification shows relevant discrimination power into the impacts, especially if analysed jointly with other factors such as seasonality and environmental conditions. I consider the analysis to be quite thorough and the presentation of the results is very clear, in particular all the graphics have been developed quite carefully.

I think this manuscript provides a relevant contribution to impact studies driven from a good understanding of the meteorology of windstorms, and could promote further studies focused on other regions affected by ETCs and other types of impacts.

We thank the reviewer for their insightful comments and are pleased to hear that they found the paper's findings relevant and the presentation clear. We have copied all detailed comments below in blue and provided our responses to each specific comment in black.

Some minor comments I think the authors should address are:

General:

Would the authors consider that there could be a benefit in redefining the large domain to reduce the influence of the O category cyclones? There is no discussion in sections 4.1 or 5.

Reply: We agree that these kinds of box-type areas are of course arbitrary. In this study, however, we wanted also to see the "bigger picture" of windstorms occurring close to Finland. Also, in Section 4.2 there is some discussion when we say that *"In the characteristic comparison, we only considered classes F because a comparison with classes O would not be fair. This is due to the large size of the bounding box including numerous cyclone tracks in class O that do not affect the domain of Finland at all."*

For figs 5 and 6, I would suggest a reconsideration for the colours, since these are nor colour-blind friendly (most common type of colour blindness can't distinguish between red and green), but even for a standard sighted person I find it hard to distinguish the different tones in Fig 6, so maybe combine with different symbols?

Reply: We have now revised the colors in all figures as "colorblind safe" using https://colorbrewer2.org/. In addition, we revised Figure 6 so that it uses symbols to better separate the classes.

Minor comments on text

   Line 8 (abstract): it currently read "northern part of a country", but it should be of THE country, as this is not generic

Reply: Yes, you are correct. We have revised accordingly.

Lines 74-76, sentence starting with "Furthermore" though this might become clearer once the reader has covered the whole manuscript, this description is very confusing so early on in the text. I suggest the authors should review it and focus on the key aspects.

Reply: We have now reviewed the text beginning with the previous sentence and simplified the "Furthermore" sentence to clarify the points. The text now reads as follows (lines 73-79): *"We address these objectives by developing a novel classification for all extratropical cyclones and windstorms, based on their arrival location and direction as well as the climatological locations of the strongest wind gusts, and by identifying windstorms through their impact (power outages) rather than solely through meteorological features. Furthermore, we compare the meteorological characteristics of windstorms to extratropical cyclones, determine windstorm-related meteorological properties (e.g., min MSLP), and quantify how the impacts vary depending on the type of windstorm and its meteorological characteristics. We further investigate how the impacts vary by region and season."*

Lines 107-108: On the issue of weather ERA5 overestimate or underestimates winds and gusts, the literature is a lot more nuanced. I suggest the authors take some time to look for example at the references below. There is specific literature that has focused on the performance for ETCs in particular:
https://rmets.onlinelibrary.wiley.com/doi/10.1002/joc.8339 --> "ERA5 shows a good skill for wind speed with normalized mean bias (NMB) of −0.7% and normalized root-mean-square error (NRMSE) of 14.3%, despite a tendency to overestimate low winds and underestimate high winds"

https://confluence.ecmwf.int/display/CKB/Windstorm+footprints%3A+Product+User+Guide --> "It was found that wind gust from reanalysis (ERA-Interim and ERA5) underestimates measured wind gust on average"

And more generally there is a distinction between performance onshore/offshore and for low winds/high winds, and with topography and land use features
https://link.springer.com/article/10.1007/s00382-020-05302-6

https://rmets.onlinelibrary.wiley.com/doi/pdf/10.1002/qj.3616

https://wes.copernicus.org/articles/9/1727/2024/

https://www.sciencedirect.com/science/article/pii/S2352484723015603

https://asr.copernicus.org/articles/17/63/2020/ --> "ERA5 is very skilled, despite its low resolution compared to the regional models, but it underestimates wind speeds, especially in mountainous areas"

https://ges.rgo.ru/jour/article/view/3328/761 --> "The assessments revealed a

systematic error at most stations; in general, ERA5 tends to overestimate wind speed over forests and underestimate it over grasslands and deserts."

 I would suggest that the authors review their statement and add a bit more detail.

Reply:  We appreciate the comment and the references and have added some additional details and references to the manuscript in lines 111-118: *"ERA5 demonstrates good performance in wind and gust analyses but has certain biases. It tends to overestimate low wind speeds while underestimating high winds, with wind gusts frequently being underestimated (Chen et al., 2024). The accuracy of ERA5 varies by region, performing better offshore than onshore and facing challenges in mountainous areas (Minola et al., 2020). Ongoing (unpublished) work at the Finnish Meteorological Institute has compared ERA5 10-m winds and wind gusts to observations and found that results in Finland are largely similar to those reported by Minola et al. (2020) for Sweden: weak winds are overestimated, while high winds and gusts are underestimated. Additionally, land cover influences wind speed estimates, with overestimations over forests and underestimations over grasslands and deserts (Shestakova et al., 2024). "*

Line 396: If the values are not normalised by number of storms, then the conclusion is not as direct as there are also more storms on autumn than winter. This  should at least be mentioned.

Reply: The normalised, i.e., the averaged NDP per windstorms, are described on lines 430-432, where it is stated that: *"If we look at the average NDP numbers per windstorm, the most damage actually occurs during the summer (188 000 NDP per windstorm). In winter the number is 166 000/windstorm, autumn: 150 000/ windstorm and spring: 94000/windstorm."*